# A histomorphological atlas of resected mesothelioma discovered by self-supervised learning from 3446 whole-slide images

Farzaneh Seyedshahi [1,2], Kai Rakovic [1,2,3,12], Nicolas Poulain[2,12], Adalberto Claudio Quiros[1,4], Ian R. Powley [2], Cathy Richards[5], Hussein Uraiby[5], Sonja Klebe [6], David A. Moore [7,8], Apostolos Nakas[5], Claire R. Wilson [9], Marco Sereno [10], Leah Officer-Jones [2], Catherine Ficken[2], Ana Teodosio[11], Fiona Ballantyne[2], Daniel Murphy [1,2], Ke Yuan [1,2,4] ✉ & John Le Quesne [1,2,3] ✉

Mesothelioma is a highly lethal and poorly biologically understood disease which presents diagnostic challenges due to its morphological complexity. This study uses self-supervised AI (Artificial Intelligence) to map the histo-morphological landscape of the disease. The resulting atlas consists of recurrent patterns identified from 3446 Hematoxylin and Eosin (H&E) stained images scanned from resected tumour slides. These patterns generate highly interpretable predictions, achieving state-of-the-art performance with 0.65 concordance index (c-index) for outcomes and 88% AUC in subtyping. Their clinical relevance is endorsed by comprehensive human pathological assessment. Furthermore, we characterise the molecular underpinnings of these diverse, meaningful, predictive patterns. Our approach both improves diagnosis and deepens our understanding of mesothelioma biology, highlighting the power of this self-learning method in clinical applications and scientific discovery.

Mesothelioma is a highly lethal cancer almost always caused by asbestos exposure[1,2]. Early detection, critical for effective treatment, remains challenging[3,4]. Mesothelioma's biological diversity complicates histopathological diagnosis, as early malignancy can be difficult to distinguish from reactive changes. Diagnosing mesothelioma from H&E (Hematoxylin and Eosin) images is a subjective and time-intensive process even for skilled subspecialty histopathologists. Definitive diagnosis often remains elusive[5,6], even with immunohistochemistry or FISH (fluorescence in situ hybridisation). These diagnostic challenges are at least in part due to the difficulties in devising robust, manually applicable systems of morphological characterisation, in addition to

the well-known issues of inter-pathologist agreement. The emergence of AI methods provides an opportunity to comprehensively describe the morphological complexity of mesothelioma to generate a quantitative visual dictionary of the disease.

Recent AI methods in mesothelioma primarily focus on tile-based or cell-based approaches, using supervised or weakly-supervised learning. MesoNet[7] used Whole Slide Images (WSI) tiles to predict patient survival through a risk score based on malignant morphologies but lacked insight into the diversity of the tumour microenvironment. Cell-based methods like MesoGraph[8] and SpindleMesoNET[9] quantified malignancy through tumour cell shapes, especially spindle cells, but

[1]School of Cancer Sciences, University of Glasgow, Glasgow, Scotland, UK. [2]Cancer Research UK Scotland Institute, Glasgow, Scotland, UK. [3]Pathology Department, Queen Elizabeth University Hospital, NHS Greater Glasgow and Clyde, Glasgow, Scotland, UK. [4]School of Computing Science, University of Glasgow, Glasgow, Scotland, UK. [5]University Hospitals of Leicester, Leicester, UK. [6]Flinders Health and Medical Research Institute, Adelaide, Australia. [7]CRUK Lung Cancer Centre of Excellence, UCL Cancer Institute, London, UK. [8]Department of Cellular Pathology, University College Hoapital, London, UK. [9]Leicester Medical School, University of Leicester, Leicester, UK. [10]University of Leicester, Leicester, UK. [11]Birmingham Tissue Analytics, University of Birmingham, Birmingham, UK. [12]These authors contributed equally: Kai Rakovic, Nicolas Poulain. ✉e-mail: ke.yuan@glasgow.ac.uk; john.lequesne@glasgow.ac.uk

required extensive annotations and were computationally intensive for slide-level applications. While these approaches provide valuable insights, their findings are restricted by the nature and quality of their human annotations. Recent self-supervised models, such as Hierarchical Image Pyramid Transformer (HIPT)[10], CTransPath[11], HistoSSLscaling[12], UNI[13], and Histomorphological Phenotype Learning(HPL)[14] as well as other self-supervised models such as RNAPath[15], which focus on healthy tissue analysis, have been developed for H&E WSI histopathological analysis. Uniquely among these approaches, HPL focuses on identifying recurrent histomorphological patterns through clusters known as histomorphological phenotype clusters (HPCs). HPL leverages the Barlow Twins self-supervised framework[16], using ResNet for feature extraction from 224 × 224 WSI patches, followed by clustering of tile feature vectors via the Leiden algorithm[17]. Each HPC represents a unique morphological pattern that can be associated with specific molecular landscapes or used to predict patient outcomes and mesothelioma subtypes by quantifying HPC frequencies. (Further details in the 'Online Methods' section.) Previously, HPL has been applied to lung cancer, revealing significant underlying patterns and giving impressive prognostic performance[14], but it has not been implemented in mesothelioma. Self-supervised methods such as HPL depend upon accessing large volumes of training data, preferably from resected tumour material which offers large tissue areas with full morphological variance. This is especially challenging in mesothelioma, which is so often diagnosed from tiny biopsies and subsequently treated medically rather than surgically.

In this work, we curated 3446 whole slide images of 485 resected mesothelioma cases to generate a uniquely powerful training resource called Leicester Archival Thoracic Tumour Investigation Cohort-Mesothelioma (LATTICe-M), as shown with further details in Fig. 1a. We then applied the HPL pipeline to our dataset to build a comprehensive atlas of mesothelioma H&E morphology. (Fig. 1b)

## Results
### Mapping the histomorphological phenotype landscape
We have identified 47 recurrent histomorphological phenotype clusters (HPCs) (Fig. 2a) based on morphological features encoded by self-learned neural networks. These HPCs are identified from 3,239,939 tiles extracted from 3446 images at 5x equivalent resolution. 41 of 47 HPCs are shared in more than 20% of cases, and none of them are case-specific. A threshold of >1% abundance was applied to call an HPC "present" in a case. HPCs were then binned by patient prevalence groups as well as coloured by rare and frequent (<20% and > 80%) in grey, intermediate (20−80%) in blue. As a result of this, two complementary bar charts summarise these distributions: one showing the percentage of cases per HPC and another counting HPCs within 10%-wide patient-prevalence bins. Rare HPCs (<20% prevalence) represent either normal tissues (open lung/muscle, which are minor tissue components in the tumour-rich blocks selected for scanning), reactive changes which are either unusual or not targeted for scanning (dense lymphocytes from tertiary lymphoid structures, pleural plaque), and a couple of the less common tumour phenotypes (cold, solid pattern epithelioid disease and plump disorganised spindle cells). The near-universal HPCs (>80%) represent features which are either very widespread in a surgical resection (e.g. talc pleurodesis, vessels, collagen) or quite broad ubiquitous malignant morphologies (e.g. infiltrated fat, sparse epithelioid disease). Interestingly, these more common HPCs often display lower 'purity', reflecting a broader morphological composition.

A team of subspecialty expert pathologists from 3 centres, who had no access to the WSI images or labels (blinded assessment), examined every HPC to achieve consensus morphological annotations for each one, derived from their defining features: epithelioid vs spindled morphology, inflammation, necrosis, cellularity, desmoplasia, atypia, and cluster purity and each HPC was given a summary title.

They evaluated inflammation levels in each HPC, categorising them as None-Sparse, Mild-Moderate, or Marked. Most HPCs were None-Sparse, but some displayed notable patterns. Assessments of inflammation and necrosis exhibited the highest levels of consensus, with at least 50% of the HPCs receiving unanimous agreement in these categories. For epithelioid growth patterns, we also observed a relatively high level of full agreement. However, for spindle architecture in our non-epithelioid clusters (orderly/less orderly/disorderly), agreement among the pathologists was lower, perhaps reflecting the subjectivity of this measure. Across 47 HPCs with 3 raters, Fleiss' Kappa scores (reported in the last row of Fig. 2c—with variable category definitions per component) for individual histopathological components ranged from 0.2 to 0.6, indicating fair to moderate agreement based on the interpretation scale proposed by ref. 18. This degree of agreement is in line with kappa scores for several diagnostic tasks in mesothelioma[19]. These annotations reveal areas of the UMAP containing multiple HPCs with broad similarities, such as spindled/collagenous HPCs, epitheloid tumour growth patterns, and lymphocytic infiltration, as well as peripheral and projecting clouds of morphologically highly distinct lung tissue and chest wall muscle tiles (Fig. 2d). This grouping is supported by the general co-occurrence of HPCs within each slide in the Supplementary Figs.. These HPCs enable the detailed automated spatial annotation of any mesothelioma whole slide image, as illustrated in Fig. 2e, highlighting two cases with highly divergent outcomes and morphologies. The first case, a sarcomatoid malignancy which resulted in death at 66 days, is highly morphologically diverse and contains abundant tiles in spindle cell-associated clusters, while the second is predominantly made up of a single epithelioid cluster.

### HPCs predict mesothelioma subtypes
The crucial histopathological distinction in mesothelioma subtyping lies between epithelioid and non-epithelioid (i.e. sarcomatoid/biphasic) variants. To classify this, we generated a numerical vector representing the percentage or frequency of each HPC for every WSI. This vector was transformed using the centred log-ratio (clr) transformation to enhance stability and interpretability, then feed into a logistic regression model for classification as either non-epithelioid or epithelioid. Tumours labelled as biphasic and sarcomatoid were combined into a single group, creating a binary classification task. 8 HPCs are significantly associated with the epithelioid subtype (HPCs 14, 39, 24, 25, 27, 40, 8, and 18), containing epithelioid malignancy, predominantly characterised by tubular patterns and solid sheets of epithelioid tumour cell growth. Of the 9 HPCs linked to the non-epithelioid subtype, 3 HPCs (15, 16, and 22, mostly containing disorderly spindle cells) are unanimously classified as non-epithelioid malignancy by our pathologists as well. The other 6 HPCs (6, 7, 35, 37, 45, and 28, mostly solid epithelioid growth pattern) contain more diverse appearances, including epithelioid HPCs, pleural plaque and muscle. This might be due to the inclusion of biphasic cases and lethal epithelioid patterns in this group and also suggests a possible link between sarcomatoid growth and invasion into the chest wall. (Fig. 3a) Our logistic regression classifier (Likelihood Ratio test statistic(40) = 1219.3, $p$ = 1.195e-229) achieved 88% 5-fold cross-validated AUC (Area Under the Curve) Score on the LATTICe-M dataset and 80% on The Cancer Genome Atlas (TCGA) mesothelioma dataset and robust across varied clustering configurations (Fig. 3b). We visualised HPC compositions at the case level using a PCA plot, colour-coding cases by subtype at diagnosis. The transition from epithelioid cases to sarcomatoid cases through biphasic cases is clearly visible. (Fig. 3c)

### HPCs as predictors of patient survival
We aggregated HPC frequencies across all samples per patient, summarising each case into a readily interpretable composition of morphologies. The 5-fold cross-validated c-index values for patient prognosis outcomes were 0.67 and 0.65 for the training and test

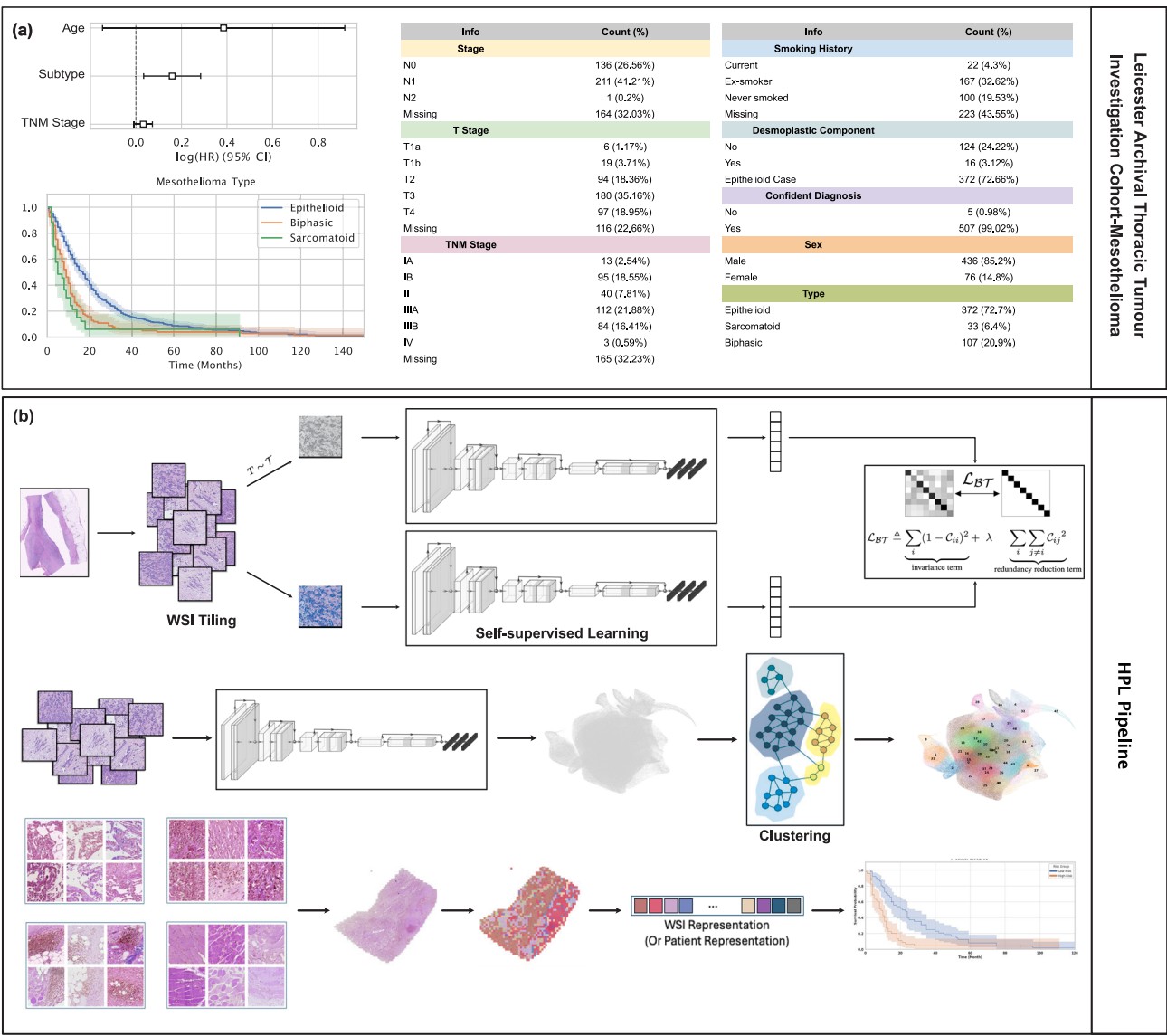

**Fig. 1 | LATTICe-M dataset overview and HPL pipeline for mesothelioma analysis. a** An overview of the LATTICe-M dataset clinical information, highlighting key demographic and pathological information. The forest plot displays log hazard ratios (centre) with 95% confidence intervals (error bars), derived from the Cox proportional hazards model for clinical variables including age, subtype, and TNM stage. Survival probability Kaplan-Meier survival curves are stratified by mesothelioma subtype, with time displayed in months. Shaded areas represent 95% confidence intervals around each survival curve. Risk groups were defined by median predicted hazard from the Cox model. The overview plots are based on $n = 512$ patients (biological replicates), each representing an independent clinical record. The unit of study is patient and no technical replicates is used. Other clinical variables are presented with their corresponding frequencies and percentages in two summary tables. **b** HPL pipeline workflow: Each WSI is divided into 224 by 224 pixel tiles. After applying various data augmentation distortions, these tiles served as input for the Barlow Twins self-supervised learning model. Once the model is trained, the ResNet backbone network generated 128-dimensional feature vectors per tile, representing prominent histopathological features. These vectors were then grouped using the Leiden community clustering algorithm to identify morphologically distinct patterns. At the patient level, the clusters representing different histopathological patterns were analysed to quantify the proportion of each HPC within each WSI. This quantitative information was subsequently used to predict mesothelioma subtypes and patient outcomes. Source data are provided as a Source Data file.

LATTICe-M primary datasets, respectively, and 0.65 for the fully unseen TCGA cohort as an external dataset. The addition of clinical information, including mesothelioma subtype, TNM stage, and age, only modestly improved the ability of the algorithm to predict outcomes, yielding an increment in C-index of 0.01. We further verified that our approach is robust across other clustering configurations. Compared to similar research on mesothelioma outcome prediction using WSIs, such as MesoNet[7], our model achieves at least a comparable c-index score for the same additional dataset (TCGA). While MesoNet reported a score of 0.656 for TCGA, we matched this performance, with c-index scores ranging from 0.64 to 0.7 across different folds, however, prioritising model interpretability through our morphology-based HPCs, also using a fully self-supervised pipeline.

We identified HPCs 10 (Log Hazard Ratio = −0.089, $p = 0.001$, Confidence Interval = [−0.145, −0.034]) and 27 (Log Hazard Ratio = −0.062, $p = 0.008$, Confidence Interval = [−0.109, −0.016]) ("epithelioid nests in bland stroma" and "dense lymphocytes") as positive survival factors, while HPCs 15 (Log Hazard Ratio = 0.052, $p = 0.026$, Confidence Interval = [0.006,0.098]) and 22 (Log Hazard Ratio = 0.042, $p = 0.016$, Confidence Interval = [0.008, 0.077]) ("disorderly spindle cells" and "transitional mesothelioma") emerge as strong

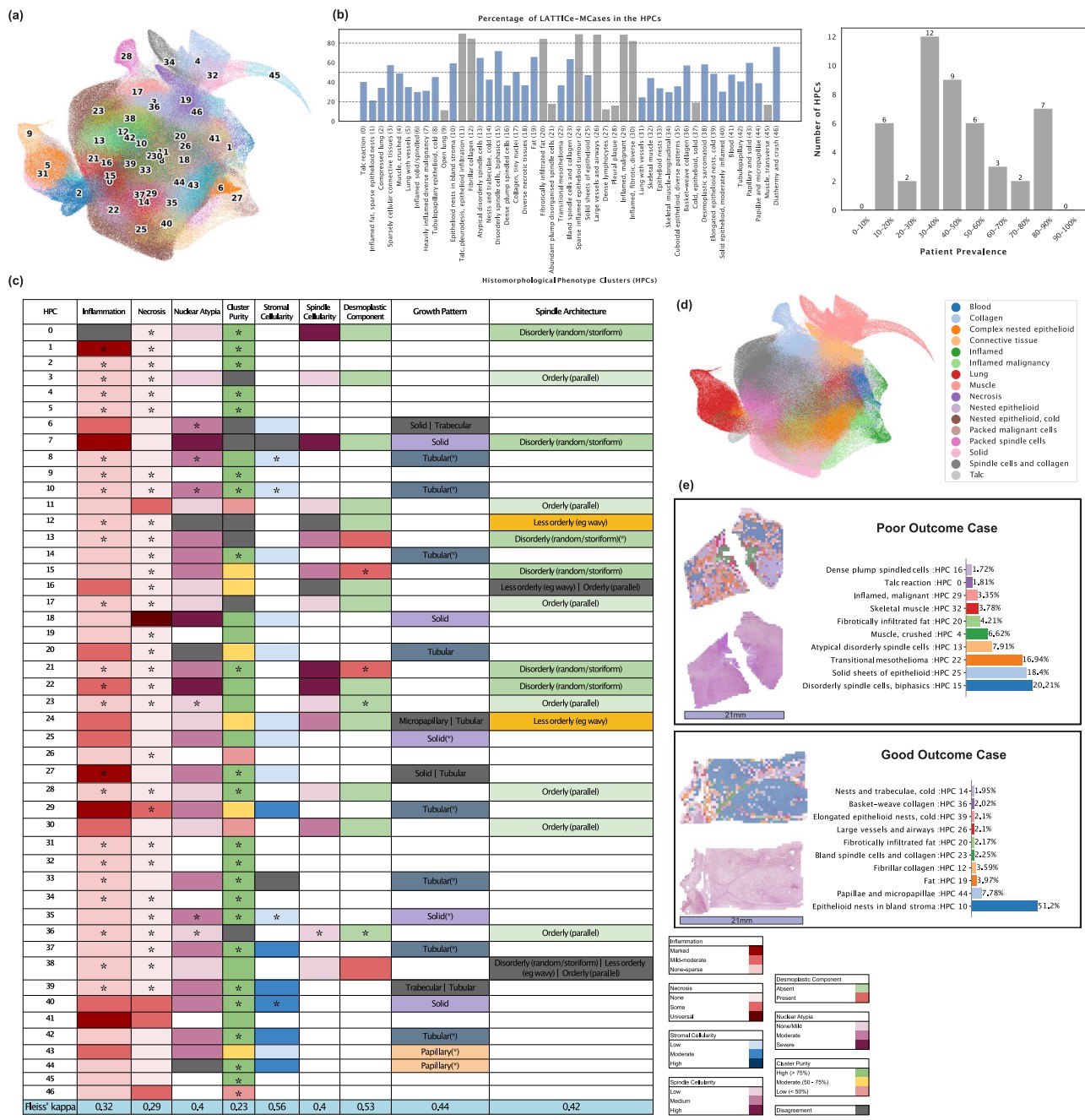

**Fig. 2 | HPC analysis and pathologist validation of mesothelioma tissue patterns. a** A HPC color-coded UMAP (Uniform Manifold Approximation and Projection) plot displaying the spatial distribution of 47 distinct HPCs. **b** Distribution and patient prevalence of HPCs: Percentage of LATTICe-M cases exhibiting each HPC, with colouring indicating rare and ubiquitous (<20% or >80%, grey) and intermediate (20−80%, blue) patterns. Also on the right, distribution of HPCs categorised by patient prevalence decile. **c** Majority consensus for pathologists' panel's annotations for HPCs. Asterisks (*) indicate complete agreement among all pathologists. **d** Main pattern color-coded UMAP, where annotations were provided by a pathologist following the clustering step to group HPCs with similar morphologies. **e** WSI from a poor outcome and good outcome case overlaid and quantified by HPCs, demonstrating the differences in HPC composition and showing the percentage of the top 10 HPCs for each sample, alongside their histomorphology annotations (scale bar, 21mm). Source data are provided as a Source Data file.

predictors of poor outcome. (Fig. 4a) A comparison of tile-level UMAP plots colour-coded by hazard ratio reveals a high degree of similarity, further underscoring the strong links between sarcomatoid transformation and poor patient outcomes. The map highlights red areas (higher hazard ratio) like HPC 15 and 16 (Sarcomatoid HPCs) and blue areas (lower hazard ratios) like HPC 45, 27 and 5 (either non-tumourous tissue or lymphocyte HPCs) (Fig. 4b).

Next, we categorised patients into high- and low-risk groups based on their calculated hazard ratios for 60 months. Kaplan-Meier

plots were generated for LATTICe-M train (Log-rank test statistic(1) = 62.41, $p$ = 2.79e-15) and test (Log-rank test statistic(1) = 15.14, $p$ = 9.96e-05) datasets, as well as the TCGA-Meso additional cohort (Log-rank test statistic(1) = 10.24, $p$ = 0.00138), as shown in Fig. 4c. The model achieves impressive separation, predicting outcomes with surprising power in this very poor-prognosis population who face all the complex hazards of radical surgery and impaired respiratory physiology alongside the biology of their tumour burden. Figure 4d shows a comparison between classical histological grading of epithelioid

(a)

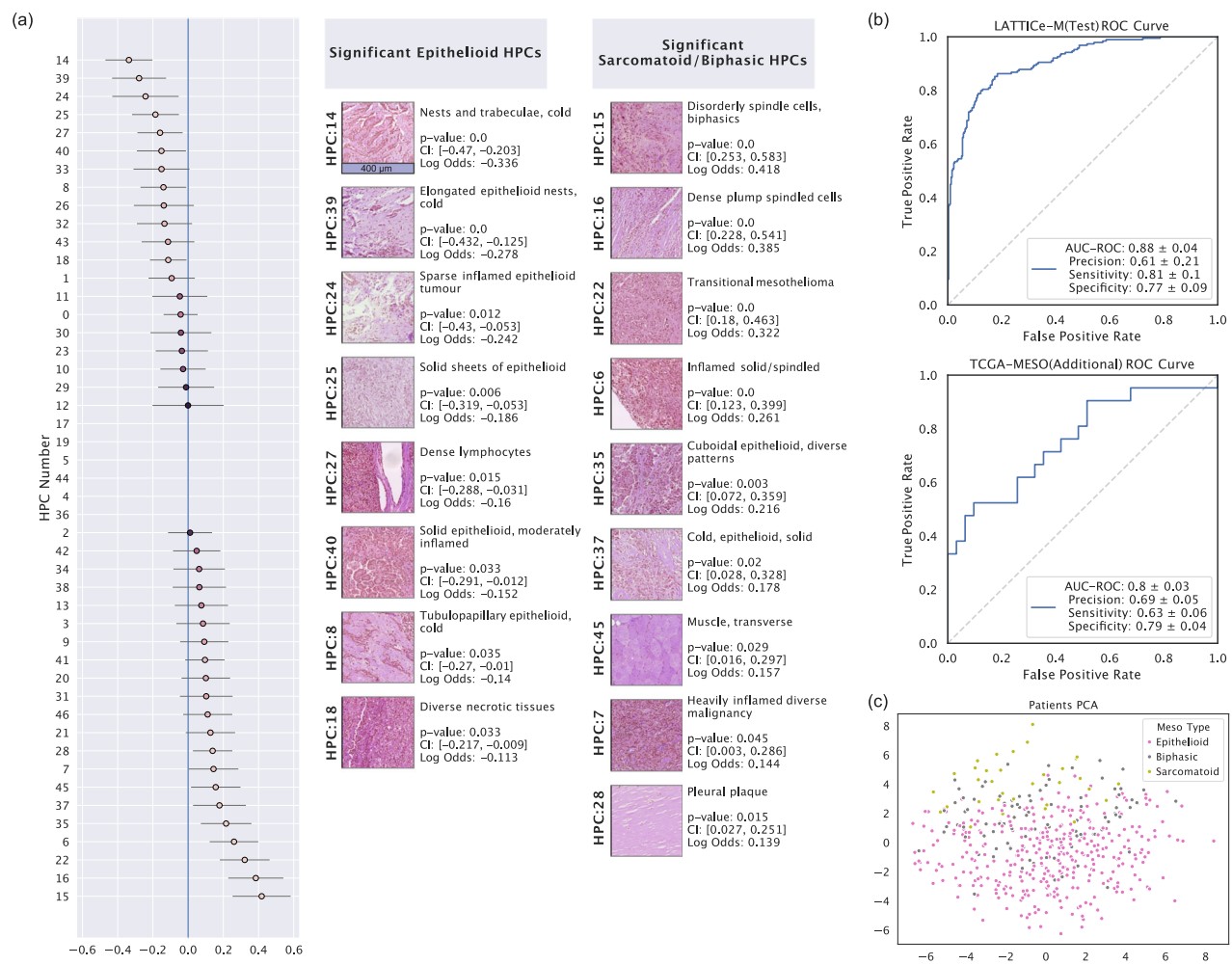

(b)

(c)

**Fig. 3 | HPC performance in mesothelioma subtype classification. a** Forest plot for the logistic regression model used in subtype classification, showing log odds ratios (centre) with 95% confidence intervals (error bars), derived from logistic regression coefficients as each HPC's contribution towards subtype prediction, along with significant HPCs for each class (epithelioid versus non-epithelioid), including their *p*-values, confidence intervals, log odds ratios, and pathologist annotations on HPC histomorphology. This analysis includes *n* = 3446 WSIs (biological replicates) with subtype labels derived from clinical data. WSIs from epithelioid and non-epithelioid cases (*n* = 2565 and *n* = 881, respectively) were treated as independent samples. No technical replicates were included (scale bar, 400 μm). **b** The ROC (Receiver Operating Characteristic) curve for LATTICe-M test and TCGA-MESO datasets showing their subtype classification performance, including AUC-ROC (Area Under the Curve for ROC), Precision, Sensitivity and Specificity scores. **c** The PCA (Principal Component Analysis) plot shows patient-level vector representations, color-coded by mesothelioma subtype labels. Source data are provided as a Source Data file.

pleural mesothelioma (as suggested in ref. 20) and our model, both applied to the TCGA mesothelioma just epithelioid sample cases. Our pipeline demonstrates superiority (Log-rank test statistic(1) = 5.02, *p* = 0.025) against human grading (Log-rank test statistic(1) = 1.16, *p* = 0.282) for patient outcomes in this dataset. Figure 4e shows the SHAP (SHapley Additive exPlanations)[21] decision plots for our Cox model, comparing a high-risk sarcomatoid case (red) and a relatively low-risk epithelioid case (blue). The plot shows how the model assigns high or low-risk labels for these patients based on the abundance/scarcity of influential HPCs, such as the highly lethal HPC 15 ("disorderly spindle cells"), or the protective HPC 27 ("dense lymphocytes"), which both contribute to the calculated risk in these two cases.

We further demonstrate the ability of our model to predict patient outcomes within disease subtypes (epithelioid vs non-epithelioid) groups. HPC frequencies were calculated, and survival was predicted separately for each group, identifying HPCs which underscored subtype-specific traits (Fig. 5a). For epithelioid cases, HPC 10 ("epithelioid nests in bland stroma") and HPC 22 ("transitional mesothelioma") emerged as significant predictors of good and bad outcomes, respectively. HPC 10 is likely to represent relatively indolent well-differentiated classically epithelioid disease. Interestingly, HPC 22, which is very enriched for the appearances of transitional mesothelioma, is not uncommon in cases diagnostically subtyped as epithelioid and is strongly predictive of poor outcomes in this group. This supports the view that transitional appearances signal early stages of transition to sarcomatoid growth[22], and its presence in this group highlights the difficulty in human identification of this pattern[23]. For biphasic/sarcomatoid cases, HPC 23 ("bland spindle cells and collagen") predicts poor outcome, perhaps identifying areas of cytologically bland desmoplastic differentiation, while the good prognostic association of HPC 27 ("dense lymphocytes") suggests towards a particular role for the immune system in sarcomatoid disease. We also show example tiles for specific HPC groups of interest based on pathologist annotations, including inflamed clusters, classical desmoplastic appearances, and necrosis (Fig. 5b).

To further assess the biological significance of the identified HPCs, we investigated their associations with quantitative Immunohistochemistry (IHC) markers reflecting tumour cell proliferation and aberrant mRNA translation activity. HPCs with significant associations to previously obtained quantitative IHC markers[24] are shown in Fig. 5c.

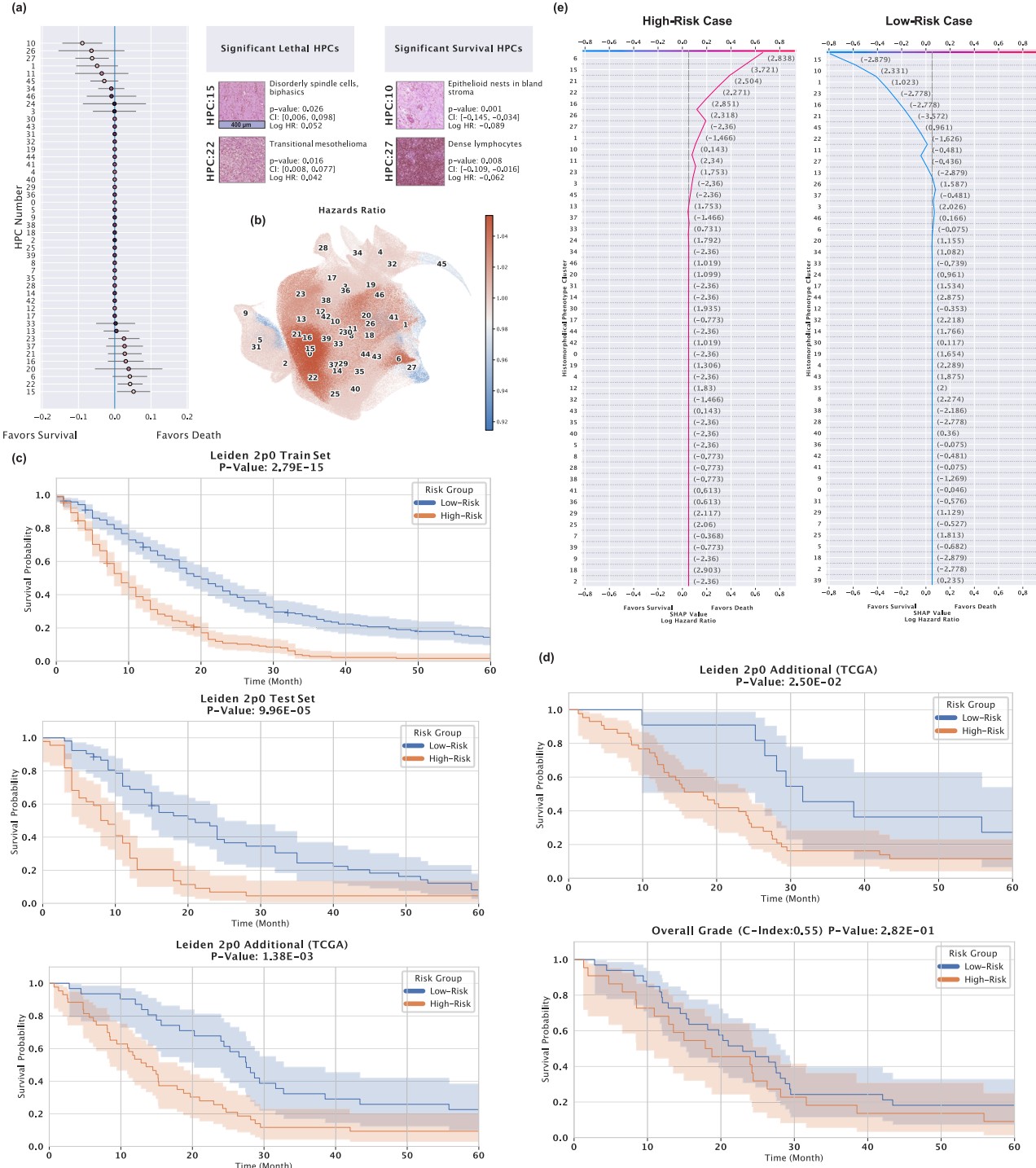

**Fig. 4 | HPC performance in mesothelioma survival prediction and risk stratification. a** Forest plot for the Cox proportional hazards model used in survival prediction and values represent log hazard ratios (centre) with 95% confidence intervals (error bars), obtained from the Cox model, including Significant HPC's $p$-values, confidence intervals, log hazard ratios, and pathologist annotations on HPC morphology. The Cox model was trained on $n = 512$ independent patients, each linked to a survival outcome. All samples are biological replicates, and the unit of study is the patient. No technical replicates were used in the analysis (scale bar, 400 μm). **b** UMAP plot of tile vector representations, color-coded by Cox model predicted hazard ratio for each HPC. **c** Kaplan-Meier plots showing low-risk and high-risk patient groups for LATTICe-M and TCGA-MESO datasets, with their reported $p$-value performance scores. Shaded bands represent 95% confidence intervals. **d** Kaplan-Meier plots comparing the performance of our survival predictor, pre-trained on the LATTICe-M dataset, tested on TCGA-MESO epithelioid cases, versus predictions based on the traditional systemic overall grade of the TCGA-MESO epithelioid cases. Shaded bands represent 95% confidence intervals. **e** An example of a SHAP decision plot for a high-risk (red line) and low-risk (blue line) patient, displaying the percentage of HPCs within their samples that contribute to their outcomes. Source data are provided as a Source Data file.

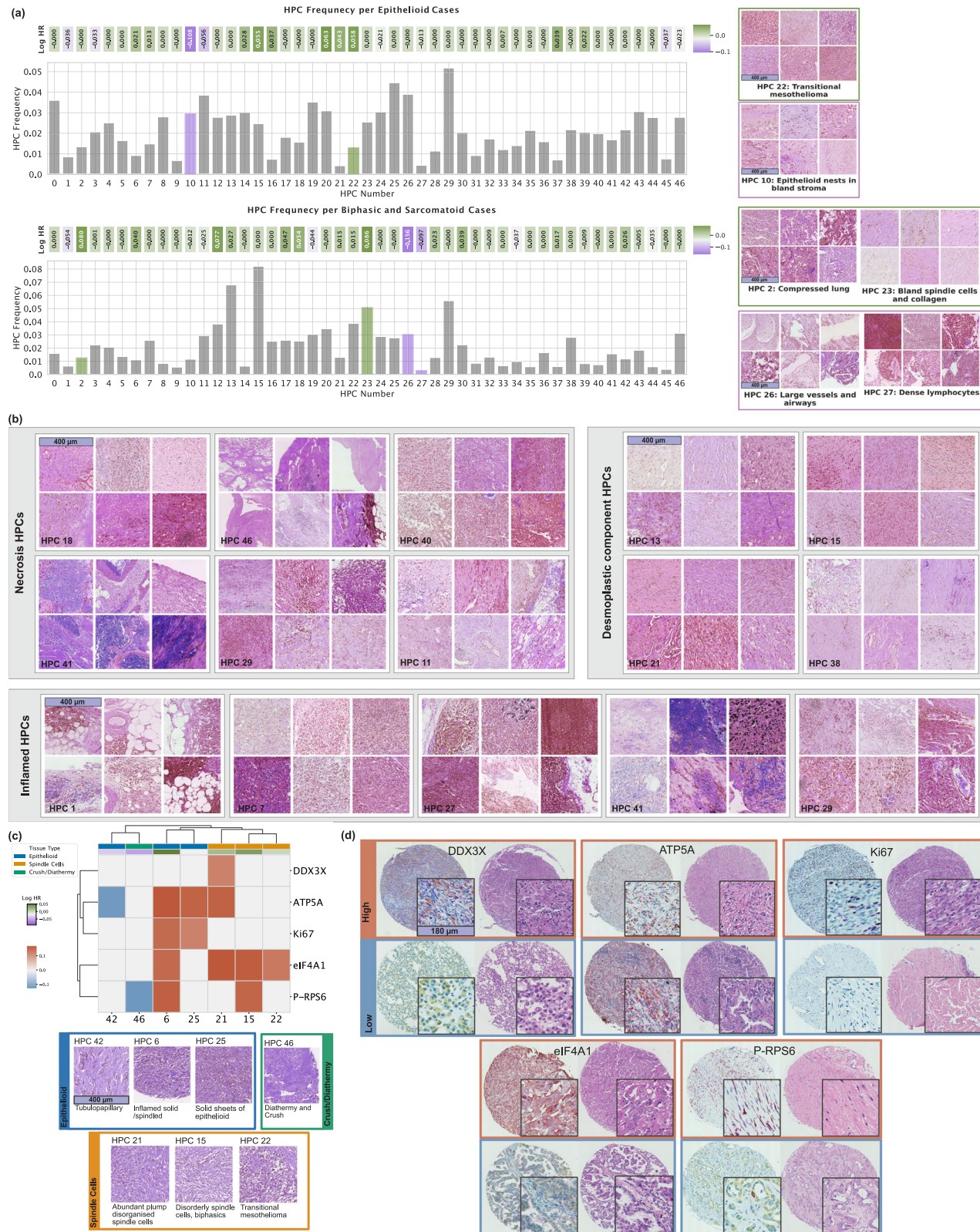

Notably, the HPCs with upregulation of mRNA translation, proliferation, and oxidative phosphorylation are nearly all associated with poor patient outcome and are all either sarcomatoid or poorly differentiated epithelioid in morphology, further underlining the linkage of these processes to tumour virulence. eIF4A1, the ubiquitous pro-proliferation translation initiation factor, is particularly closely related to poor outcome HPCs, supporting possible therapeutic targeting of this molecule. Negative associations with markers of oxidative phosphorylation and pro-translation mTOR signalling are only seen in areas of low-grade disease, or crush/diathermy artefact likely to degrade IHC signal. Figure 5d represents chromogenically IHC-stained tissue cores for each marker. The top row shows examples with high expression of

**Fig. 5 | Subtype-specific HPC analysis and immunohistochemical marker correlations. a** HPC frequency plot for epithelioid (n = 372) and biphasic/sarcomatoid (*n* = 140) cases, showing the co-occurrence of HPCs over cases within each group. Log hazard ratios are drawn based on a Cox model trained on subtype-filtered cases, 5-fold cross validated, and significant HPCs (*p* < 0.05) are highlighted in bold colors. **b** Grouped HPCs based on pathologist annotations, highlighting key histopathological features such as necrosis, desmoplastic components, and inflammation in the tumour microenvironment. 6 random tiles are displayed for each

HPC. **c** Correlations between HPC proportions and positivity for IHC markers per TMA cores (*n* = 711) were assessed using two-sided Spearman rank correlation tests. Positive correlations (red colors) indicate that the marker is enriched in that HPC; negative correlations (blue colors) indicate depletion. Only HPCs with significant adjusted *p*-values, alongside associated HPCs, are shown. **d** Representative IHC-stained cores showing high expression (top row) and low expression (bottom row) for associated IHC markers (scale bars, 400 μm (**a–c**), 180 μm (**d**)). Source data are provided as a Source Data file.

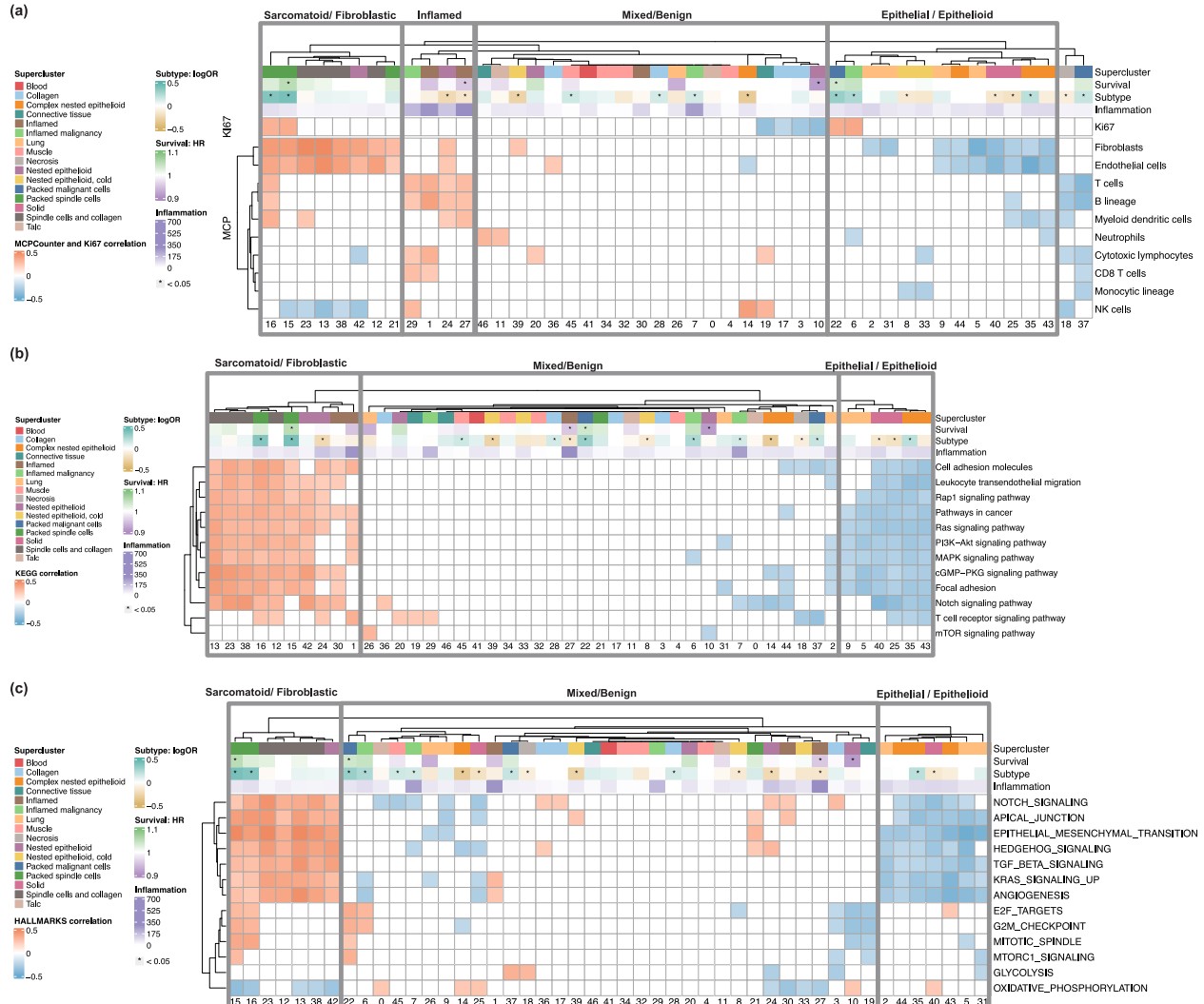

**Fig. 6 | Correlation between WSI-level HPC compositions and transcriptomic signatures in TCGA.** Two-sided Pearson correlation between WSI level HPC compositions and transcriptomic signatures in bulk RNA-seq data from the TCGA-MESO cohort. Only correlation coefficients with associated *p*-value lower than 0.05 are represented; supercluster annotations were provided by an expert pathologist. The subtype row shows the log odds ratio from the logistic regression model, while

survival row is based on the partial hazards ratio predicted by the Cox model. The inflammation row was generated using inference from the HoVer-Net model. **a** Ki67 (Proliferation Marker) expression and MCPcounter (Microenvironment Cell Populations-counter) tumour microenvironment signatures. **b** KEGG (Kyoto Encyclopedia of Genes and Genomes) pathway signatures. **c** MSigDB (Molecular Signatures Database) hallmark gene sets. Source data are provided as a Source Data file.

the corresponding marker, while the bottom row shows cores with low expression. For each case, both the IHC-stained image and the corresponding H&E scan are displayed side by side. Additionally, a representative tile from each core is shown to highlight the cellular-level resolution of the tissue.

## Molecular underpinnings of HPCs
We next investigated the biological underpinnings of HPC morphology to further explain our model's predictive capabilities in mesothelioma

prognosis and subtyping. This was achieved by quantifying associations between gene expression signatures and HPC composition in the TCGA Mesothelioma RNASeq dataset[25].

We used the MCPcounter algorithm to estimate cell types, including fibroblasts, endothelial cells, T cells, B lineage cells, myeloid dendritic cells, NK cells, and CD8 T cells from RNASeq data. (Fig. 6a) Expression of the proliferation marker Ki67 is also mapped, revealing especially high proliferation in spindle cell-enriched HPCs (HPCs 22, 16, 15, and 6). Critically, these HPCs are the most predictive of non-

epithelioid subtype (3a). In contrast, HPCs defined by well-differentiated epithelioid disease and normal tissue (e.g., HPCs 17, 3, 10, and 19) show low proliferation.

Fibroblast signatures are strongly pronounced in multiple clusters which either significantly determine sarcomatoid disease or contain fibroblastic/collagenous/stroma-rich morphology indicative of fibroblast-like mesenchymal dedifferentiation of mesothelioma cells. Fibroblast signatures are minimal in HPCs representing papillary or micropapillary epithelioid mesothelioma, solid pattern disease, lung tissue, and large-vessel-rich HPCs consistent with more specialised epithelioid/tissue-specific phenotypes.

Lymphocyte-rich HPC 27 shows strong correlations with T cells, B lineage cells, and myeloid dendritic cells. Similarly, inflamed HPCs (HPCs 29, 1, 24, 27), identified by Hover-Net, indicate active immune environments linked to better prognosis and likely improved immunotherapy response. HPCs 1 (inflamed fat) and 27 (dense lymphocytes) are high in both B- and T-cell signatures and show strong inter-correlation, suggesting dense inflammation and tertiary lymphoid structure formation in chest wall fatty tissues, supporting previous observations that tertiary lymphoid structures are related to good outcome.[26]

KEGG pathway correlations across HPCs again show clear separation between non-epithelioid and epithelioid subtypes (Fig. 6b), with generally heightened mitogenic signalling pathway activity in a group of sarcomatoid and fibroblastic clusters associated with aggressive biology. In contrast, HPCs linked to epithelioid growth and normal tissues exhibit relative down-regulation.

An analysis of cancer hallmark pathways further identifies the most sarcomatoid HPCs as a group with strong positive links to multiple proliferation-associated pathways (Fig. 6c), in addition to mitogenic signalling and multiple EMT-related pathways. Notably, the same group exhibits downregulation of oxidative phosphorylation components, indicating a metabolic shift towards hypoxia.

### Validation on tiny tissue fragments

To assess the generalisability of our self-supervised model trained on the LATTICe-M dataset, we benchmarked its performance on the St. George's Hospital TMA dataset from the MesoGraph study[8]. This external evaluation is significant for two reasons. First, no additional training was applied to the new dataset, so the results represent the pre-trained model's performance on a fully unseen wholly exterior cohort. Second, although the model was trained on WSIs, it maintained strong performance on tissue microarray cores. These fragments are not only tiny ( ≈ 1 millimetre) but are selected to represent pure tumour tissue. In contrast, our model was trained on large diagnostic WSIs including background tissues, and used unsupervised clustering to filter artefacts.

As the image size in TMA cores is insufficient to support the previous frequency-based method, we employed multiple instance learning (MIL) to predict mesothelioma subtypes by summarising information across tiles from a core or a biopsy. We called this method HPL-MIL and benchmarked HPL-MIL against state-of-the-art methods, max-MIL and naive-MIL (patch-based MIL methods), PINS[27], CLAM[28], MesoGraph, on the 235 cores from St. George's Hospital TMA cohort[8] (Table 1). Each TMA core was treated as a bag of instances, where the instances are individual tile embeddings extracted from the core. Using an attention-based multiple instance learning approach, we obtained a core-level representation by computing a weighted average of tile embeddings. We then performed subtype classification of each core using the core-level labels available for the TMA dataset. HPL-MIL achieved higher AUC, Average Precision, Sensitivity and specificity scores across all the methods without pre-training on the cohort tissues.

### Discussion

In this study, we applied our self-supervised HPL pipeline to the LATTICe-M cohort, which we believe to be the largest image collection

in terms of area of mesothelioma tissue yet employed for AI training. We achieved state-of-the-art accuracy in two key clinically important tasks: a C-index of 0.65 in survival prediction across a 5-fold cross-validation and 88% AUC in subtype classification (epithelioid vs sarcomatoid/biphasic). Furthermore, our method outperformed human grading in prognostication for epithelioid cases, and we identified survival-linked histomorphological patterns within each subtype, emphasising the interpretability of self-supervised methods and identifying recurrent morphologies worthy of future study. Quantitative visual maps of HPCs (Fig. 2e) and SHAP decision plots (Fig. 4e) offer clinical utility for understanding AI diagnoses and future selection of therapies.

Our approach eliminates the need for retraining to retain performance on external mesothelioma datasets, thus addressing a key computational challenge in self-supervised models, as proven by its efficacy across three independent cohorts. It effectively extracts relevant morphological patterns from small TMA cores (e.g. the St. George's dataset) and WSIs of varied origins and quality (e.g. TCGA and LATTICe-M), enabling real-time clinical decision-making without extensive preprocessing. CLAM[28] was benchmarked against HPL on both the TCGA and LATTICe-M datasets (full results in Supplementary Data). HPL consistently outperformed CLAM in both subtype classification and survival prediction, while maintaining high interpretability and biological relevance. This suggests the possibility of a robust diagnostic tool for both resection material and mesothelioma biopsies, which remain a major diagnostic challenge.

Our model has essentially created a morphological atlas of mesothelioma, discovering ab initio the characteristic recurrent H&E morphologies which comprise the disease. The fact that these morphologies have clear biological and clinicopathological significance proves their meaning and value. For example, the discovered linkage of the tumour microenvironment to patient survival shows how crucial the morphology of immune system engagement is to tumour virulence and biology, and suggests biomarker potential in predicting responses to immunotherapy. Furthermore, RNASeq data annotation of histomorphological clusters further illustrates connections between tumour microenvironment signatures, molecular pathways, and survival, offering valuable molecular insights into the biology of the disease.

The molecular associations of HPCs help us to understand tumour virulence and suggest numerous hypotheses for mechanistic testing. For example, we see numerous mRNA cancer hallmark pathways linked to high-risk sarcomatoid HPCs, helping to explain links between morphology and outcome in molecular terms and highlighting possible areas of target discovery. Sarcomatoid clusters are directly linked to signatures of proliferation, hypoxia, and EMT in bulk sequence data, without any requirement for spatial methods or microdissection. This is in keeping with biological knowledge that sarcomatoid mesothelioma cells can proliferate rapidly under hypoxic conditions[29] and supportive of the idea that sarcomatoid dedifferentiation represents co-option of a physiological EMT pathway.

Additionally, subtle transitional morphologies in cases classified as being epithelioid overall appear to have significant prognostic value in our survival analysis. This highlights the continuous nature of epithelioid to sarcomatoid transition, and suggests the importance of accurate identification of transitional states, which is a challenging task by eye, and which is likely to benefit from our approach. Furthermore, HPL could also be used to target therapy by identifying such cases with subtle sarcomatoid changes, which are likely to be more responsive to immunotherapies[30].

This study also has several limitations that warrant acknowledgment. First, staging data were missing in 32.23% of Leicester cases (165 patients), probably reducing the power of T/N/M-related analysis (Fig. 1a). Second, smoking history was incomplete in 43.55% of cases (223 patients), limiting cohort-wide assessment of its impact. These

**Table 1 | Performance inference metrics for HPL with Multiple Instance Learning (MIL) and other MIL-based approaches over TMA Core dataset**

| Method | AUC-ROC | Avg. Precision | Sensitivity | Specificity |
|---|---|---|---|---|
| max-MIL | 0.70 ± 0.01 | 0.54 ± 0.12 | 0.54 ± 0.07 | 73 ± 0.09 |
| naive-MIL | 0.84 ± 0.05 | 0.72 ± 0.11 | 0.72 ± 0.08 | 0.71 ± 0.1 |
| PINS | 0.85 ± 0.05 | 0.80 ± 0.07 | 0.82 ± 0.1 | 0.71 ± 0.13 |
| CLAM | 0.85 ± 0.07 | 0.74 ± 0.11 | 0.75 ± 0.11 | 0.77 ± 0.02 |
| MesoGraph | 0.90 ± 0.007 | 0.86 ± 0.02 | 0.88 ± 0.015 | 0.72 ± 0.01 |
| HPL-MIL (Ours) | 0.93 ± 0.12 | 0.92 ± 0.11 | 0.94 ± 0.09 | 0.86 ± 0.19 |

AUC-ROC (Area Under the Curve for ROC), Average Precision, Sensitivity, and Specificity scores for the HPL using MIL approach to generate unique risk/sarcomatoid-ness scores. The St. George's Hospital TMA core dataset was used for benchmarking without any prior training, as reported in ref. 8, through internal cross-validation, demonstrating the robustness of tile vector representations from our self-supervised network trained on mesothelioma WSIs.

data gaps highlight the need for consistent clinical documentation in retrospective studies and constrain the use of these variables in survival and subtype prediction models alongside AI-derived features (HPC frequencies).

## Methods

### Datasets

The primary dataset used in this study is the Leicester Archival Thoracic Tumour Investigation Cohort-Mesothelioma (LATTICe-M)[31], comprising 512 patients diagnosed with pleural mesothelioma who underwent surgical resection. Study clinical data were collected and managed using REDCap electronic data capture tools[32,33] hosted on secure research servers at University Hospitals of Leicester NHS Trust. Cases are histologically subtyped into epithelioid ($n = 372$), sarcomatoid ($n = 107$), and biphasic ($n = 33$). The cohort includes 436 male and 76 female patients (85.2% and 14.8%, respectively), consistent with mesothelioma incidence at the collection site. Sex was self-reported at the time of intake. Patient age ranged from 36 to 85 years (64.3 ± 8.6). No information on race, ethnicity, or other socially relevant variables was collected. Participants were not financially compensated. Sex and gender were reported in the study; however, no sex-based analysis was performed with the aim of training a self-supervised model. Disaggregated sex counts are available in the source data files.

Ethical approval was obtained from the UK National Health Service Research Ethics Committee (ref. no. 14/EM/1159). No prospective recruitment, interventions, or international data transfers were involved. There were no risks to participants or researchers, as only archived histopathology material was used under standard governance. Pathology annotation support was provided by S.K. (Adelaide, Australia), whose contributions were formally recognised through authorship. Data ownership is held by the Greater Glasgow and Clyde Biorepository, under governance via an amendment granted by the Leicester South REC. All research procedures were conducted in compliance with relevant ethical regulations, and written informed consent was obtained from all participants.

Figure 1 a presents additional clinical details. The WSIs were sectioned and stained with Hematoxylin and Eosin at Leicester University Hospital, scanned at 10X, 20X, or 40X magnifications. After tiling and background removal, WSIs with fewer than 100 tiles were excluded, leaving 485 patients and 3446 WSIs for the pipeline and downstream analysis. To identify significant clinical factors in this cohort, we employed a Cox proportional hazards model and found that age, mesothelioma subtype, and TNM stage significantly contributed to survival prediction. (Fig. 1a)

To validate our results, we used the publicly available Cancer Genome Atlas (TCGA)-mesothelioma cohort[25], an entirely differently-scanned dataset, still comprising WSIs but obtained from multiple centres. It includes 86 samples from 74 patients with both WSIs and RNAseq data available. This cohort was primarily used to discover links between HPCs and tumour microenvironment features, pathways, and

hallmarks. All HPL pipeline steps were performed on the primary dataset (LATTICe-M), and evaluation scores were reported on the fully unseen additional TCGA dataset, without any further training.

Finally, we utilised the St. George's Hospital dataset, consisting of H&E-stained TMAs from tumour biopsies collected at St. George's Hospital, London. This dataset includes four TMA slides scanned at 20x magnification using a Hamamatsu Nanozoomer S360 scanner, comprising 235 cores labelled as epithelioid, biphasic, or sarcomatoid, as the only available clinical information. The dataset, introduced in the Mesograph study[8], was used for training and testing. We employed it to demonstrate the robustness and generalisability of our trained WSI model by benchmarking and comparing its performance on TMA cores against different methods reported in the study. (Section 3)

### Histomorphological phenotype learning (HPL)

HPL is a tool developed to detect and categorise histomorphological patterns within large collections of whole-slide images. HPL employs an automated, self-supervised deep learning approach, eliminating the need for expert pathologists to prelabel or manually define histomorphological patterns. Once these patterns are identified, new whole-slide images can be introduced to the trained model and classified according to the pre-established patterns. This feature allows pathologists to quantify specific patterns in new patient samples precisely. The clustering of each whole-slide image into meaningful histomorphological patterns follows several sequential steps, which are described below (Fig. 1b)

- *Whole-slide images pre-processing:* In this first step, whole-slide images are segmented into non-overlapping 224 × 224-pixel tiles at 5X magnification, which corresponds to a pixel size of approximately 1.8 micrometres. Tiles that do not contain at least 60% tissue coverage are filtered out to maintain relevance. Consistent pixel size and magnification are ensured during the tile processing phase to guarantee uniformity in the resulting tiles. The tiling code used for this process is accessible on DeepPATH GitHub[34] for further details.

- *Feature extraction:* HPL employs a self-supervised learning technique known as Barlow Twins[16], which matches or even exceeds the performance of other self-supervised methods. Barlow Twins delivers state-of-the-art results in standard pathology tasks compared to DINO[35], MoCo[36], and SwAV[37] methods[38]. We previously compared Barlow Twins with DINO within HPL framework, and it showed improved performance with the cohort size similar to this study[14]. One key feature of HPL is its ability to maintain consistent image representations, even with slight colour or zoom level variations. This capability ensures that differences in image scanning or processing across datasets do not affect the results. The aim is to capture diverse visual patterns in tissue samples and represent them as feature vectors, capturing distinct characteristics like texture. Each 224 × 224-pixel tile is

converted into a vector representation, denoted as $\{z \in R^D; D = 128\}$. During training, the model is optimised to produce consistent outputs for twin inputs, ensuring robustness in vector representation.

- *Clustering:* After generating the vector representations, we employed the Leiden community detection algorithm[17] (from the Python ScanPy library[39]) to cluster tiles or vector representations with similar histomorphological features. Since neighbouring vector representations in high-dimensional space exhibit similarity, this method effectively groups the tiles based on shared morphological patterns captured by their feature vector representations.

  We began with a subsample of 750,000 tiles and constructed a nearest-neighbour graph between the tiles. From this initial set of detected clusters, we assigned the remaining vector representations to these clusters (or graph nodes) based on their distance. The number of clusters identified depends on the chosen Leiden algorithm resolution. For our analysis, multiple resolutions were applied to capture varying levels of granularity. The resulting histomorphological phenotype clusters (HPCs) enabled the quantification of patients or whole-slide images (WSIs) based on these clusters, streamlining further analysis and simplifying the understanding of complex tissue patterns.

- *Preparing compositional vectors:* At this stage, using the identified HPCs, we can characterise the entire tissue or patient by quantifying the frequency of each HPC (1). To achieve this, each whole-slide image (WSI) is transformed into a compositional vector **A**, where the dimensionality is equal to the total number of HPCs ($c$). Each element within the vector represents the percentage of the tissue area attributed to a specific HPC. This approach quantifies the contribution of each HPC to the overall tissue composition, allowing for a detailed analysis of the histomorphological landscape within a patient or a sample.

$$A = \{a_0, a_1, a_2, \cdots, a_{c-1}\} \text{ s.t. } \sum_{i=0}^{c-1} a_i = 1 \text{ and } a_i \in [0, 1] \tag{1}$$

For statistical compositional analysis and to prepare for the use of linear models, we apply the Centred Log-Ratio (clr) transformation[40] to our compositional vector **A** to minimise correlation between HPC frequencies. This transformation maps the vector composition from the $c$-part simplex into a $c$-dimensional Euclidean vector space. Additionally, to address zero elements in the dataset, we use multiplicative replacement[41].

- *Subtype classification:* For the diagnostic task, we employed the clr transformed compositional vectors derived from whole-slide images (WSI) and fed them into a logistic regression model (Scikit-learn[42] and Statsmodels Python library[43]). This approach is weakly-supervised, utilising patient-level labels assigned by pathologists, where each patient label is applied to both the patient and their corresponding slides. We combined sarcomatoid and biphasic mesothelioma into a single class (non-epithelioid class) and compared it against the majority class, primarily consisting of epithelioid samples. Also, to address the class imbalance in the primary dataset (1:3 ratio for non-epithelioid to Epithelioid subtypes), we applied an undersampling strategy using the Edited Nearest Neighbour (ENN) technique[44] (Imbalanced-learn Python library[45]). This method reduced the majority class by removing redundant and noisy samples.

  Ultimately, the logistic regression model used the compositional vectors of WSIs to classify mesothelioma subtypes based on the contributions of HPCs. In this approach, individual HPCs serve as distinct features for our logistic regression classifier, enabling us to rank the importance of each HPC and its role in predicting specific tumour subtypes within each sample. The predicted probability of being two classes is given by:

$$Y = \frac{e^{b_0 + b_1 * clr(A)}}{1 + e^{b_0 + b_1 * clr(A)}} \tag{2}$$

Where $b_0$ is the bias or intercept term and $b_1$ is the coefficient for compositional vector ($A$).

- *Survival analysis:* In the clinical outcome aspect of our study, we created a clr-transformed compositional vector for each patient, reflecting the overall HPCs composition. We then used the Cox proportional hazards regression model[46] to analyse patient survival in relation to the HPC composition vector. Finally, Kaplan-Meier plots[47] were employed to visually distinguish between high-risk and low-risk patient groups within each dataset. For this step, we used Lifelines[48] and SciPy Python libraries[49]. For both subtype classification and survival prediction tasks, we employed five-fold cross-validation to ensure robust evaluation. The reported scores represent the average performance across all folds. Furthermore, we ensured no overlap of patients between the training and test sets, maintaining strict separation to prevent data leakage and guarantee unbiased assessments. However, for providing annotations and associations with the tumour microenvironment, we focused on a single fold for consistency and detailed analysis.

## Expert pathologist annotation

Additionally, we engaged three expert pathologists to independently and blindly annotate HPCs without access to patient clinical data or additional HPC details. Each HPC was classified into one of three categories: epithelioid tumour, spindle cells/extracellular matrix, or non-tumour. The pathologists assessed each HPC's primary and secondary architectural features, HPC purity, inflammation, necrosis, nuclear atypia and biphasic components (in malignant groups). Also, they evaluated patterns such as desmoplasia and cellularity in spindle cell HPCs, as well as the tumour-stroma ratio and stromal cellularity in epithelioid HPCs.

To assess agreement in our multi-centre annotation process, we used majority voting among the three expert pathologists who annotated the HPCs. Instances of unanimous agreement, where all three pathologists selected the same category and are marked with an asterisk (*). In contrast, cases of complete disagreement, where each pathologist chose a different category, are highlighted in grey in Fig. 2c. Also, inter-rater reliability was assessed using Fleiss' Kappa. For each HPC $i$ ($N = 47$), we counted the number of raters ($n = 3$) assigning it to each category $j$, and computed the marginal probability of category $j$ as:

$$p_j = \frac{1}{Nn} \sum_{i=1}^{N} n_{ij}, \qquad 1 = \sum_{j=1}^{k} p_j \tag{3}$$

We then calculated the average proportion of agreeing rater-pairs across clusters (observed agreement $\bar{P}$), estimated the agreement expected by chance and scaled the excess agreement relative to the maximum possible beyond chance:

$$\kappa = \frac{\bar{P} - \bar{P}_e}{1 - \bar{P}_e} \tag{4}$$

A Kappa of 1 indicates perfect concordance, 0 reflects agreement no better than random, and values ≤0 denote worse-than-chance

agreement. It is important to note that the number of categories available for annotation (denoted as $j$) varied across the different histomorphological components we selected, such as inflammation, necrosis, etc. As a result, the Fleiss' Kappa values are inherently influenced by this variability and are not directly comparable across components. Specifically, components with more annotation categories introduce greater choice complexity, which tends to lower agreement scores. To prevent misinterpretation, we recommend referring to the majority voting results and the asterisk indicators of full agreement as complementary measures of reliability.

## HPC cell type enrichment analysis

We used the deep learning model HoVer-Net[50] to segment cells in each tile within the HPCs and calculate the abundance of only inflammatory cells in every HPC. While the tiles used in the HPL framework were at 5x magnification, the HoVer-Net model was trained on 20x tiles. To bridge this difference, we first applied HoVer-Net to 20x tiles, then combined 16 tiles (arranged in a $4 \times 4$ grid) to create 5x equivalents. This allowed us to map HoVer-Net's segmentation results, particularly the inflammation annotations, to specific tiles. Each tile was subsequently assigned to a corresponding HPC by calculating the average number of inflamed cells detected across the related tiles. Approximately 900 whole-slide images (WSIs) at 20x magnification were annotated using the HoVer-Net model for this analysis.

## HPC tumour microenvironment signature associations

We correlated HPCs with tumour microenvironment features, hallmark pathways, and relevant biomarkers (Section 3). All WSIs from each patient were used to calculate the clr-transformed HPC compositional vectors. We then applied the single-sample Gene Set Enrichment Analysis (ssGSEA) to quantify pathway expression in both the Kyoto Encyclopedia of Genes and Genomes (KEGG)[51] and Molecular Signatures Database (MSigDB)[52] hallmark datasets. We also estimated immune cell subpopulation abundance using MCPcounter[53]. Ki67 RNA expression levels were also utilised across the entire sample set to assess cellular proliferation in each case, providing further insight into tumour growth activity within different morphological HPCs. Correlations between the clr-transformed HPC compositions and pathway expression levels were calculated, with only correlations having a $p$-value below 0.01 retained, ensuring statistical significance.

To further evaluate the biological relevance of the identified HPCs, we sought associations between HPCs and quantitative IHC measures of tumour cell proliferation and dysregulation of mRNA translation. We used data previously generated from a study of the LATTICe-M TMA cohort which revealed the importance of translational dysregulation to mesothelioma development[24]. Data were available from 8 TMAs, comprising 711 cores after quality control. To link molecular phenotype with spatial composition, we calculated the proportional representation of each HPC within each TMA core and then assessed the association between HPC proportions and marker expression. Marker positivity scores were derived from automated quantification pipelines applied to scanned IHC images. The correlation was calculated using the two-sided Pearson test and multiple comparison correction was applied using the Benjamini-Hochberg false discovery rate (FDR) method with $\alpha = 0.05$.

## Tissue microarray benchmarking

We also benchmarked and compared the HPL model (trained on WSIs) against other state-of-the-art AI methods using an independent small dataset of tissue microarray (TMA) cores to demonstrate its robustness. We used the St. George's Hospital dataset, publicly released with the MesoGraph study[8], comprising 235 cores with associated mesothelioma subtype labels. The study benchmarked methods such as max-MIL, naive-MIL (patch-based MIL approaches), PINS[27], CLAM[28], and MesoGraph on this dataset. Employing gated attention in MIL[54], we

predicted the probability of each core belonging to a specific mesothelioma subtype, naming this approach HPL-MIL. In our weakly-supervised MIL setting, each TMA core is treated as a bag $B = \{h_1, h_2, \ldots, h_k\}$ of $k$ tile embeddings (Instances). Each tile $h_k$ is obtained from our HPL ResNet-128 encoder trained using the Barlow Twins framework on the LATTICe-M dataset. To derive a representation for the entire core, we use attention-based pooling:

$$\mathbf{z} = \sum_{k=1}^{K} a_k \mathbf{h}_k \tag{5}$$

where the attention weight $a_k$ is computed as:

$$a_k = \frac{\exp\left\{\mathbf{w}^\top \tanh\left(\mathbf{Vh}_k^\top\right)\right\}}{\sum_{j=1}^{K} \exp\left\{\mathbf{w}^\top \tanh\left(\mathbf{Vh}_j^\top\right)\right\}} \tag{6}$$

This allows the model to learn which tiles are more informative for core-level prediction. The resulting representation $z$ is passed to a linear classifier for subtype classification. While MIL lacks full interpretability, it allowed us to benchmark mesothelioma subtype classification against existing MIL-based methods, representing the generalisability and scalability of the HPL pipeline. Despite this success, the remainder of the study prioritises the more interpretable histomorphological clusters and their morphological insights to address the complexity of mesothelioma.

We additionally benchmarked the CLAM (Clustering-constrained Attention Multiple Instance Learning)[28] framework using 128-dimensional tile embeddings extracted from our Barlow Twins-trained ResNet. Subtype classification was performed using a linear layer on top of CLAM outputs, while survival prediction was based on risk scores generated by the network and evaluated via a Cox proportional hazards model, enabling a fair comparison with HPL-based survival predictions. CLAM was trained for 50 epochs with early stopping, using the Adam optimiser with a binary loss and a learning rate of $10^{-4}$. The total loss combined slide and instance-level objectives with coefficients $c_1 = 0.9$ and $c_2 = 0.3$, as follows:

$$Loss_{total} = c_1 \times Loss_{slide} + c_2 \times Loss_{tile} \tag{7}$$

The number of clusters was fixed at 8, consistent with the original CLAM configuration. WSIs were treated as bags, with subtype labels assigned at the bag level, and a gated attention, was used to compute instance-level attention. Five-fold cross-validation, aligned with the HPL evaluation, was applied throughout.

## Reporting summary

Further information on research design is available in the Nature Portfolio Reporting Summary linked to this article.

## Data availability

The LATTICe cohort (histology whole slide images and clinical data) used in this study is not publicly available due to their extremely large size and ethical limitations according to the LATTICe agreement, which makes public hosting technically impractical. However, we are delighted to make the data available for academic research purposes upon request. Interested researchers may contact the corresponding author via the email provided. Access will be granted for a limited period based on a clear research purpose and mutual agreement, with data use restricted to non-commercial research. We aim to respond to access requests as soon as possible. TCGA mesothelioma RNAseq data have been retrieved from UCSC Xena [https://xenabrowser.net/datapages/?cohort=GDC%20TCGA%20Mesothelioma%20(MESO)] and images from Genomic Data Commons (GDC) portal [https://www.cancer.gov/ccg/research/genome-sequencing/tcga/studied-cancers/

mesothelioma-study]. St. George Hospital TMA Dataset is available on MesoGraph GitHub [https://github.com/measty/MesoGraph]. Source data are provided with this paper.

## Code availability

The source code for this study is openly available under the MIT License on GitHub [https://github.com/FarzanehSeyedshahi/Histomorphological-Phenotype-Learning] and archived on Zenodo[55]. Reproducible figures can be generated using the provided Jupyter notebooks. A step-by-step README on GitHub details installation and execution.

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

## Acknowledgements

The authors would like to express their gratitude to the CRUK Scotland Institute, the NHS Greater Glasgow and Clyde Biorepository, the University of Leicester, and the University of Glasgow for their invaluable support and contributions to this study. The manuscript was critically reviewed by Catherine Winchester (CRUK Scotland Institute). CRUK Early Detection Programme, IAMMED-Meso, EDDPGM-Nov21\100001 supported the research. J.L.Q. is supported by the Mazumdar-Shaw Molecular Pathology Chair endowment at the University of Glasgow. KY acknowledges support from Cancer Research UK (EDDPGM-Nov21\100001 and DRCMDP-Nov23\100010), BBSRC BB\V016067\1, Prostate Cancer UK MA-TIA22-001 and EU Horizon 2020 grant ID: 101016851.

## Author contributions

F.S. conceived the study, conducted the research, performed data analysis, and wrote the manuscript. K.R. provided pathology expertise, insights, and contributed to TCGA data annotations. N.P. performed R codings and TCGA sequencing data associations. A.C.Q. provided computational guidance insights for HPL analysis. C.R., S.K., and J.L.Q. performed histopathological annotations and provided mesothelioma subspecialty biological expertise. K.Y., D.M., and J.L.Q. supervised the project, provided guidance throughout, and contributed insights for running the project. I.R.P., H.U., D.A.M., A.N., C.W., M.S., L.O., C.F., A.T., and F.B. contributed to LATTICe-M dataset curation. All authors reviewed and approved the final manuscript.

## Competing interests

D.A.M. has received speaker fees from AstraZeneca, Eli Lilly, BMS, Takeda and Boehringer Ingelheim; consultancy fees from AstraZeneca, ThermoFisher, Takeda, Amgen, Janssen, MIM software, Bristol-Myers Squibb and Eli Lilly; and educational support from Takeda and Amgen. All other authors declare no competing interests.
