## [Peer Review File · Nature Communications]

A histomorphological atlas of resected mesothelioma discovered by self-supervised learning from 3446 whole-slide images

Corresponding Author: Professor John Le Quesne

Version 0:

Reviewer comments:

Reviewer #1

(Remarks to the Author)

Dr. Seyedshahi and colleagues have created a histomorphological atlas of mesothelioma. This is a robust project that is well designed and clearly described in this manuscript. The validation is strong and the findings are very provocative. I have the following specific comments:

- 1) Please provide references or data to support the statement that “definitive diagnosis often remains elusive”.
- 2) Page 3, line 106 – “sarcomotiod” should be “sarcomatoid”
- 3) Page 7, line 303 – “plural” should be “pleural”
- 4) The absence of stage data for about 1/3 of the patients in the Leicester data set may limit why stage/T/N/M was not significantly influential. Consider adding this limitation to the discussion.
- 5) There are similar limitations for the absent smoking history and desmoplastic data. Please comment in the discussion.
- 6) What limited the ability to assess desmoplastic component?

(Remarks on code availability)

I did not install and run the code. That is outside of my expertise.

Reviewer #2

(Remarks to the Author)

This is a comprehensive study using self-supervised AI to map the histomorphological landscape of mesothelioma. They study dataset is quite large with 3446 whole slide images from 485 patients and also used multiple datasets such as TCGA and St. George's TMA. This study is very innovative and the results may have strong impact for clinical use. There are a few issues need to be addressed before a clear conclusion can be drawn.

1. There is no external validation for the HPC grouping. Although pathologists reviewed and annotated HPC clusters, the authors did not provide the exact inter observer agreement metric to quantify how consistently different experts interpret these clusters. A quantitative measure of inter-pathologist agreement on HPC annotations will strengthen the confidence regarding the findings.

2. The model first achieved 87% AUC on the LATTICe-M dataset, but then the AUC reduced to 76.5% on the TCGA dataset. This indicate that there might be some levels of overfitting to the training data, which may due to the dataset-specific artifacts. The study did not explore techniques like domain generalization or adversarial training in order to increase external validation performance.

3. The survival prediction model has a C-index of 0.65 which is modest, and it suggest the prediction power is limited. Traditional clinical markers such as staging and other evaluation methods in combination may offer better prognostic predictions accuracy. Therefore, the authors need to take into consideration that this AI driven analysis may need further improvement in order to add clinical use value.

4. The authors should consider to conduct some IHC validation to prove the HPC-linked molecular markers which can further provide some evidence of the biological significance.

Moreover, since mesothelioma diagnosis majority is based on small biopsies instead of relatively large surgical resections. There, the dataset the authors used is mainly resected tumor tissues which may not represent what is normally encountered clinically. Therefore, this may limit the future application using this model to early stage tumor diagnosis and evaluation using small biopsies.

(Remarks on code availability)

Reviewer #3

(Remarks to the Author)

The authors apply a previously published method, Histomorphological Phenotype Learning (HPL) to Mesothelioma surgical resections (n = 3446). The authors demonstrate that they are able to predict outcomes (c-index = 0.65) and subtype cases (AUC = 85%). Using these histological clusters, they perform substantial pathological interpretation.

Overall the manuscript is well written and presents interesting findings, I have several comments I would like addressing.

1. Intro - In terms of self-supervised models mentioned in "Main" they omit several methods, including RNAPath (Cisternino et al., 2024) which is probably the closest method to HPL that's published (see Figure 2 & 3) - The difference is that method maps representations to directly interpretable substructures.

2. The authors state that 42 out of 47 HPCs are shared between more than 20% of cases. It would be nice to have a figure on the break down of this? I assume there's some HPCs in very few cases, and some in many? It would be good to understand what those extremes represent.

3. There is no information on how much expert agreement there was in terms of the following:

"A team of subspecialty expert pathologists from 3 centres, who had no access to the WSI images or labels (blinded assessment), 63 examined every HPC to achieve consensus morphological annotations for each one..."

It would be important to include agreement rates across the 3 centres to demonstrate how clean or noisy this labelling process is/was.

4. HPC proportions as a predictor of subtypes and outcomes. A good baseline would be a MIL model (e.g. CLAM) just acting on tiles of the resections. What would be the predictive accuracy of such a model? It would motivate the *need* to use HPC. The interpretation comes from the pathologists labelling anyway, so technically interpretation of the highly attended MIL tiles would result in the same thing. I see the authors did this for the TMA cores - but that section is poorly explained and it's not clear what the bag of instances are for the MIL model.

(Remarks on code availability)

Version 1:

Reviewer comments:

Reviewer #1

(Remarks to the Author)

This is an important and thoughtful analysis of mesothelioma and creates a valuable resource for researchers and scientists. My prior comments have been adequately addressed and I have no further specific comments.

(Remarks on code availability)

Reviewer #2

(Remarks to the Author)

The authors have addressed the reviewers' questions and comments. The paper has been improved significantly. There is no more question.

(Remarks on code availability)

Reviewer #3

(Remarks to the Author)

I am happy with the authors responses and I believe this manuscript warrants publication in Nature Communications

(Remarks on code availability)

RESPONSES TO REVIEWERS' COMMENTS

Manuscript reference number: NCOMMS-25-00877-T

Title: A histomorphological atlas of resected mesothelioma discovered by self-supervised learning from 3446 whole-slide images

Reviewer #1 (Remarks to the Author):

Dr. Seyedshahi and colleagues have created a histomorphological atlas of mesothelioma. This is a robust project that is well designed and clearly described in this manuscript. The validation is strong and the findings are very provocative. I have the following specific comments:

1) Please provide references or data to support the statement that “definitive diagnosis often remains elusive”.

We are happy to clarify this point. We are referring to the degree of disagreement between subspecialty experts, and now include an additional high-impact reference that highlights the diagnostic complexities of mesothelioma: Galateau-Salle et al. This study underscores the well-recognised challenges in accurately diagnosing mesothelioma.

2) Page 3, line 106 – “sarcomotiod” should be “sarcomatoid”

Thank you so much for spotting this typo. We have now corrected “sarcomotiod” to “sarcomatoid.”

3) Page 7, line 303 – “plural” should be “pleural”

Thanks again for catching this mistake. We have now corrected “plural” to “pleural”.

4) The absence of stage data for about 1/3 of the patients in the Leicester data set may limit why stage/T/N/M was not significantly influential. Consider adding this limitation to the discussion.

We appreciate your cautious interpretation and valuable suggestion. In response now, we have explicitly stated that missing staging may reduce the statistical power of evaluating the prognostic value of T, N, and M classifications in downstream predictions. The following text has been added to the discussion on page 6, line 235:

This study also has several limitations that warrant acknowledgment. First, staging data were missing in 32.23% of Leicester cases (165 patients), probably reducing the power of T/N/M-related analysis (Figure1a).

5) There are similar limitations for the absent smoking history and desmoplastic data. Please comment in the discussion.

These are also excellent points, and they prompted us to re-examine our dataset. The smoking is highly incomplete, due to the retrospective nature of the cohort. So we have added the following text to the discussion part on page 6, line 236:

Second, smoking history was incomplete in 43.55% of cases (223 patients), limiting cohort-wide assessment of its impact. These data gaps highlight the need for consistent clinical documentation in retrospective studies and constrain the use of these variables in survival and subtype prediction models alongside AI-derived features (HPC frequencies).

However, we misinterpreted the desmoplasia variable, and by including the non-sarcomatoid cases as non-desmoplastic by definition, we actually have full completeness for this key variable. Hence, the label previously marked as 'missing' for the desmoplasia component in Figure 1 has been updated to 'Epithelioid Case,' indicating that these samples are epithelioid by nature and, therefore, lack desmoplastic differentiation.

6) What limited the ability to assess desmoplastic component?

As mentioned above, the figure represented a misinterpretation of a 'missing' data tag in our epithelioid cases- which are non-desmoplastic subtype by definition and we now have changed the label in Figure1, as below:

Desmoplastic Component	
No	124 (24.22%)
Yes	16 (3.12%)
Epithelioid Case	372 (72.66%)

Reviewer #2 (Remarks to the Author):

This is a comprehensive study using self-supervised AI to map the histomorphological landscape of mesothelioma. They study dataset is quite large with 3446 whole slide images from 485 patients and also used multiple datasets such as TCGA and St. George's TMA. This study is very innovative and the results may have strong impact for clinical use. There are a few issues need to be addressed before a clear conclusion can be drawn.

1. There is no external validation for the HPC grouping. Although pathologists reviewed and annotated HPC clusters, the authors did not provide the exact inter observer agreement metric to quantify how consistently different experts interpret these clusters. A quantitative measure of inter-pathologist agreement on HPC annotations will strengthen the confidence regarding the findings.

We sincerely appreciate the reviewer's thoughtful comment regarding the consistency and transparency of our multi-centre annotation process. To address this, we updated Figure 2c to clarify the two inter-rater agreement measures. Majority voting was used, with an asterisk (*) marking full agreement (all three pathologists selected the same category), and grey shading indicating complete disagreement. Inter-rater agreement is now quantitatively assessed using Fleiss' Kappa, which extends Cohen's Kappa to multiple raters. These updates are detailed in on page 2, line 78:

Assessments of inflammation and necrosis exhibited the highest levels of consensus, with at least 50% of the HPCs receiving unanimous agreement in these categories. For epithelioid growth patterns, we also observed a relatively high level of full agreement. However, for spindle architecture in our non-epithelioid clusters (orderly/less orderly/disorderly), agreement among the pathologists was lower, perhaps reflecting the subjectivity of this measure. Across 47 HPCs with 3 raters, Fleiss' Kappa scores (reported in the last row of Figure 2c - with variable category definitions per component) for individual histopathological components ranged from 0.2 to 0.6, indicating fair to moderate agreement based on the interpretation scale proposed by [18]. This degree of agreement is in line with kappa scores for several diagnostic tasks in mesothelioma [19].

The formal definition and computation details for Fleiss' Kappa have been added to the Methods section on page 7, line 309:

To assess agreement in our multi-centre annotation process, we used majority voting among the three expert pathologists who annotated the HPCs. Instances of unanimous agreement, where all three pathologists selected the same category and are marked with an asterisk (*). In contrast, cases of complete disagreement, where each pathologist chose a different category, are highlighted in grey in Figure 2c.

Also, inter-rater reliability was assessed using Fleiss' Kappa. For each HPC i ($N = 47$), we counted the number of raters ($n = 3$) assigning it to each category j , and computed the marginal probability of category j as:

$$p_j = \frac{1}{Nn} \sum_{i=1}^N n_{ij}, \quad 1 = \sum_{j=1}^k p_j$$

We then calculated the average proportion of agreeing rater-pairs across clusters (observed agreement \bar{P}), estimated the agreement expected by chance and scaled the excess agreement relative to the maximum possible beyond chance:

$$\kappa = \frac{\bar{P} - \bar{P}_e}{1 - \bar{P}_e}$$

A Kappa of 1 indicates perfect concordance, 0 reflects agreement no better than random, and values ≤ 0 denote worse-than-chance agreement. It is important to note that the number of categories available for annotation (denoted as j) varied across the different histomorphological components we selected, such as inflammation, necrosis, etc. As a result, the Fleiss' Kappa values are inherently influenced by this variability and are not directly comparable across components. Specifically, components with more annotation categories introduce greater choice complexity, which tends to lower agreement scores. To prevent misinterpretation, we recommend referring to the majority voting results and the asterisk indicators of full agreement as complementary measures of reliability.

2. The model first achieved 87% AUC on the LATTICe-M dataset, but then the AUC reduced to 76.5% on the TCGA dataset. This indicate that there might be some levels of overfitting to the training data, which may due to the dataset-specific artifacts.

The study did not explore techniques like domain generalization or adversarial training in order to increase external validation performance.

We appreciate the reviewer for raising this point. The reasons for the drop from 87% on LATTICe-M to 76.5% on TCGA are hard to identify definitively, but will reflect fundamental differences in cohort composition, slide preparation and size difference as well as any possible overfitting issues. The LATTICe-M training dataset is a comparatively very large (512 cases, 3446 WSIs) single-center mesothelioma continuous case cohort with uniform staining and scanning protocols, whereas TCGA aggregates a much smaller set of multi-institutional samples (75 cases, 86 WSIs), and exhibits far greater variability in staining intensity, image quality and noise. Given TCGA's small size, extensive fine-tuning, adversarial training, or heavy domain-generalisation would risk overfitting and also conflict with our goal of a truly zero-effort, out-of-the-box pipeline. Hence, we retained exclusive encoder training on LATTICe-M and preserved slide-level classification as a transparent logistic model over HPC frequencies. This preserves interpretability for clinical end-users and avoids dataset-specific tailoring.

Nevertheless, in response to the reviewer's concern, and to address the performance gap without altering the core pipeline, we added a sampling step that balances class representation by removing ambiguous neighbour samples (Edited Nearest Neighbors) prior to classification, alongside a convex optimisation L1-regularised logistic classifier that directly penalises spurious feature weights.

These adjustments subtly reshape the training distribution and constrain model complexity, yielding a reliable boost in external dataset AUC (as well as primary data) without retraining or fine-tuning the backbone, and we applied the improved ROC-AUC curves in figure 3b and in the text, as below:

Finally, we should note that the Leiden clustering resolution (and thus the number of HPCs per WSI) also affects generalisation and classifier performance. While we fixed resolution at 2.0 to limit HPC counts for pathologist annotation, a higher resolution of 4.0 yields superior performance scores and a smaller domain gap, outperforming the zero-shot TCGA mesothelioma benchmark of 0.81 reported in recent work (Smith et al., Commun Biol 2025). We will include this benchmarking in the revised supplementary table 2 and hope that these additions address the concerns regarding overfitting, while preserving the simplicity and transferability of our platform.

P.S. Benchmarking of the CLAM method has also been added to this table in response to a benchmarking request raised by another reviewer.

Leiden Resolution	Dataset	Balanced Accuracy	AUC	Sensitivity	Specificity
2.0	LATTICe-M	0.79 ± 0.05	0.88 ± 0.04	0.81 ± 0.1	0.77 ± 0.09
4.0		0.79 ± 0.04	0.89 ± 0.03	0.81 ± 0.1	0.77 ± 0.06
7.0		0.79 ± 0.04	0.88 ± 0.04	0.78 ± 0.06	0.79 ± 0.08
CLAM		0.79 ± 0.03	0.87 ± 0.04	0.77 ± 0.09	0.80 ± 0.05
2.0	TCGA-MESO	0.71 ± 0.03	0.8 ± 0.03	0.63 ± 0.06	0.79 ± 0.04
4.0		0.78 ± 0.05	0.85 ± 0.02	0.7 ± 0.05	0.85 ± 0.06
7.0		0.78 ± 0.05	0.85 ± 0.04	0.67 ± 0.08	0.89 ± 0.04
CLAM		0.67 ± 0.05	0.74 ± 0.01	0.55 ± 0.04	0.79 ± 0.03

Supplementary Table 2: Subtype classification performance metrics across clustering resolutions and datasets. Balanced Accuracy, Area under the curve (AUC), Sensitivity, Specificity scores for the mesothelioma subtype classification task on LATTICe-M and TCGA-MESO cohorts across different Leiden clustering resolutions and CLAM MIL method. The scores are averaged over 5-fold cross-validation and after sampling using the edited nearest neighbour model (ENN).

3. The survival prediction model has a C-index of 0.65 which is modest, and it suggest the prediction power is limited. Traditional clinical markers such as staging and other evaluation methods in combination may offer better prognostic predictions accuracy. Therefore, the authors need to take into consideration that this AI driven analysis may need further improvement in order to add clinical use value.

We thank the reviewer for this point, and agree that a concordance of 0.65 is modest at face value. However, we do strongly feel that this is a remarkably good performance from image data alone.

By way of illustration, when we fit a combined Cox model incorporating patient age, tumor stage, and histopathological subtype alongside the self-supervised image features (Supplementary Table 3), the resulting concordance index rises only marginally from 0.65 to 0.67, indicating that these traditional markers contribute little additional prognostic signal beyond what the H&E-derived representations already capture.

The point here is perhaps that mesothelioma prognosis from morphology alone is notoriously challenging, as inter-observer agreement among expert pathologists on survival-related grading is poor, and even established nomograms seldom exceed a C-index of 0.70 in multi-institutional cohorts (link). This population of patients face complex morbidities and extensive disease, not to mention (in our training cohort at least) the impact of extremely radical surgery, so that any prognostic power from examination of small tissue fragments is, we think, quite impressive.

In this context, achieving a 0.65 C-index purely from histological images, without manual annotation or external clinical guidance, demonstrates the genuine potential of self-supervised learning in difficult prognostic tasks. We have expanded the Discussion (Section 4.2) to emphasise these findings and to note that, while modest, our C-index represents a meaningful baseline for an image-only approach. Future integration of multi-modal data (e.g., genomic profiles, radiology) may well further improve performance, and we agree is likely to be of future clinical value, but was beyond the scope of this study. The relevant clinical variables available in this cohort (including age, TNM stage, and subtype), along with the HPC-derived predictive features, are benchmarked for their prognostic value and corresponding C-indices in Supplementary Table 3 and are summarized below:

Leiden Resolutions	HPCs			Clinical and HPCs	
	Train	Test	TCGA	Train	Test
2.0	0.67 ± 0.0	0.65 ± 0.03	0.65 ± 0.01	0.68 ± 0.01	0.66 ± 0.03
4.0	0.69 ± 0.0	0.66 ± 0.03	0.66 ± 0.02	0.7 ± 0.0	0.66 ± 0.03
7.0	0.7 ± 0.01	0.65 ± 0.04	0.66 ± 0.01	0.71 ± 0.01	0.66 ± 0.04
	Risk Score			Clinical and Risk Score	
CLAM	0.60±0.01	0.60±0.04	0.61±0.0	0.62±0.02	0.62±0.05

Supplementary Table 3: Concordance indices for patient outcome prediction across datasets and clustering resolutions. Patient outcome prediction was performed using the Cox proportional hazards model, reporting concordance indices (C-indices) across train, test (LATTICE-M), and external validation (TCGA) datasets at different Leiden clustering resolutions, with 5-fold cross-validation. For the primary dataset, clinical variables (including subtype, TNM stage, and age) were integrated into the patient vectors derived from HPL. Additionally, survival analysis was independently conducted using the CLAM MIL-based approach, where risk scores were generated via gated attention mechanisms.

4. The authors should consider to conduct some IHC validation to prove the HPC-linked molecular markers which can further provide some evidence of the biological significance.

We thank the reviewer for this important point, and we fully agree that immunohistochemical evidence would strengthen the demonstration of HPC significance. Fortunately we already hold IHC data that links translational dysregulation to mesothelioma pathogenesis (link). Accordingly, we have added the following text to the revised manuscript:

This text has now been incorporated into the Results section on page 4, line 147:

To further assess the biological significance of the identified HPCs, we investigated their associations with quantitative Immunohistochemistry (IHC) markers reflecting tumour cell proliferation and aberrant mRNA translation activity. HPCs with significant associations to previously obtained quantitative IHC markers (link) are shown in Figure 5c. Notably, the HPCs with upregulation of mRNA translation, proliferation, and oxidative phosphorylation are nearly all associated with poor patient outcome and are all either sarcomatoid or poorly differentiated epithelioid in morphology, further underlining the linkage of these processes to tumour virulence. eIF4A1, the ubiquitous pro-proliferation translation initiation factor, is particularly closely related to poor outcome HPCs, supporting possible therapeutic targeting of this molecule. Negative association with markers of oxidative phosphorylation and pro-translation mTOR signalling is seen with areas of low-grade disease, or crush/diathermy artefact likely to degrade IHC signal. Figure 5d represents chromogenically IHC-stained tissue cores for each marker. The top row shows examples with high expression of the corresponding marker, while the bottom row shows cores with low expression. For each case, both the IHC-stained image and the corresponding H&E scan are displayed side by side. Additionally, a representative tile from each core is shown to highlight the cellular-level resolution of the tissue.

Also, the procedural steps are outlined in the Methods section on page 8, line 339:

To further evaluate the biological relevance of the identified HPCs, we sought associations between HPCs and quantitative IHC measures of tumour cell proliferation and dysregulation of mRNA translation. We used data previously generated from a study of the LATTICE-M TMA cohort which revealed the importance of translational dysregulation to mesothelioma development (link). Data were available from 8 TMAs, comprising 711 cores after quality control. To link molecular phenotype with spatial composition, we calculated the proportional representation of each HPC within each TMA core and then assessed the association between HPC proportions and marker expression. Marker positivity scores were derived from automated quantification pipelines applied to scanned IHC images.

Figure 5 also has been updated with the IHC Cores and the correlations:

Figure 5: Subtype-specific HPC analysis and immunohistochemical marker correlations. **a** HPC frequency plot for biphasic/sarcomatoid and epithelioid cases, showing the co-occurrence of HPCs for each subtype. Based on a Cox model trained on subtype-filtered cases, significant HPCs are highlighted in bold colors with respect to their log hazard ratio. **b** Grouped HPCs based on pathologist annotations, highlighting key histopathological features such as necrosis, desmoplastic components, and inflammation in the tumor microenvironment. **c** Heatmap showing correlations between HPC proportions and average core-level positivity for IHC markers. Positive correlations (red colors) indicate that the marker is enriched in that HPC; negative correlations (blue colors) indicate depletion. Only HPCs with significant adjusted p-values, alongside associated HPCs, are shown. **d** Representative IHC-stained cores showing high expression (top row) and low expression (bottom row) for associated markers: EIF4A1, P-RPS6, ATP5A, and Ki-67. Insets magnify representative tissue areas for cellular resolution.

Moreover, since mesothelioma diagnosis majority is based on small biopsies instead of relatively large surgical resections. There, the dataset the authors used is mainly resected tumor tissues which may not represent what is normally encountered clinically. Therefore, this may limit the future application using this model to early stage tumor diagnosis and evaluation using small biopsies.

We appreciate the reviewer's insightful note. Indeed, the majority of clinical diagnoses in mesothelioma are based on small core biopsies rather than full resections. However, by using resected tumour material we aimed to maximise the volume of training data to build a robust diagnostic framework based on complete morphological context. This provided us with the ability to explore intra-tumoral heterogeneity more comprehensively and establish a proof-of-concept for subtype prediction and survival estimation which would have been much more difficult if we had restricted our training to small biopsies.

That said, we recognise this as a potential limitation for direct clinical translation, especially in early-stage diagnoses where only small biopsies are available. To address this, our future work will include model testing and fine-tuning on small biopsy samples, which we are currently collecting and it will help us better assess the model's generalisability and ensure applicability in standard clinical workflows.

Reviewer #3 (Remarks to the Author):

The authors apply a previously published method, Histomorphological Phenotype Learning (HPL) to Mesothelioma surgical resections (n = 3446). The authors demonstrate that they are able to

predict outcomes (c-index = 0.65) and subtype cases (AUC = 85%). Using these histological clusters, they perform substantial pathological interpretation.

Overall the manuscript is well written and presents interesting findings, I have several comments I would like addressing.

1. Intro - In terms of self-supervised models mentioned in "Main" they omit several methods, including RNAPath (Cisternino et al., 2024) which is probably the closest method to HPL that's published (see Figure 2 & 3) - The difference is that method maps representations to directly interpretable substructures.

We do appreciate the reviewer for bringing RNAPath (Cisternino et al., 2024) to our attention. We agree that it is closely related to our method, as both approaches aim to extract interpretable substructures from histological data using self-supervised learning. We have now cited RNAPath in the introduction and clarified that it focuses on healthy tissue analysis. The revised sentence on page 2, line 43 reads:

UNI [13], and Histomorphological Phenotype Learning (HPL) [14], as well as other self-supervised models such as RNAPath [15], which focus on healthy tissue analysis

2. The authors state that 42 out of 47 HPCs are shared between more than 20% of cases. It would be nice to have a figure on the break down of this? I assume there's some HPCs in very few cases, and some in many? It would be good to understand what those extremes represent.

We thank the reviewer for this helpful suggestion. To better illustrate the distribution of HPCs across cases, we have revised Figure 2b by incorporating a second bar chart that visualises patient-level prevalence of each HPC. This new addition complements the existing bar plot, which shows the percentage of LATTICE-M cases assigned to each HPC.

Additionally, we enhanced the original bar plot by applying distinct coloring to indicate rare (<20%) and frequent (>80%) clusters with grey shading. This visual distinction highlights HPCs with extreme prevalence rates, allowing easier identification of outliers and biologically interesting patterns. From the updated figure, several observations can be made and reflected in the results part accordingly.

We have also expanded the Results section on page 2, line 63 to:

A threshold of >1% abundance was applied to call an HPC “present” in a case. HPCs were then binned by patient prevalence groups as well as coloured by rare and frequent (<20% and >80%) in grey, intermediate (20–80%) in blue. As a result of this, two complementary bar charts summarise these distributions: one showing the percentage of cases per HPC and another counting HPCs within 10%-wide patient-prevalence bins.

Rare HPCs (<20% prevalence) represent either normal tissues (open lung/muscle, which are minor tissue components in the tumour-rich blocks selected for scanning), reactive changes which are either unusual or not targeted for scanning (dense lymphocytes from tertiary lymphoid structures, pleural plaque), and a couple of the less common tumour phenotypes (cold, solid pattern epithelioid disease and plump disorganised spindle cells). The near-universal HPCs (>80%) represent features which are either extremely widespread in a surgical resection (e.g. talc pleurodesis, vessels, collagen) or quite broad ubiquitous malignant morphologies (e.g. infiltrated fat, sparse epithelioid disease). Interestingly, these more common HPCs often display lower ‘purity’, reflecting a broader morphological composition.

3. There is no information on how much expert agreement there was in terms of the following:

"A team of subspecialty expert pathologists from 3 centres, who had no access to the WSI images or labels (blinded assessment), 63 examined every HPC to achieve consensus morphological annotations for each one..."

It would be important to include agreement rates across the 3 centres to demonstrate how clean or noisy this labelling process is/was.

We sincerely appreciate the reviewer's thoughtful comment regarding the consistency and transparency of our multi-centre annotation process. To address this, Figure 2c has been revised to improve the interpretation of the two inter-rater agreement metrics. We applied majority voting, where full agreement among all three pathologists is now marked with an asterisk (*), and tiles with complete disagreement or uncertainty (where each pathologist selected a different category) are shaded in grey. To quantitatively capture inter-rater consistency, we calculated Fleiss' Kappa, which generalises Cohen's Kappa for use with multiple (more than 2) raters. These revisions are incorporated into the Results section on page 2, line 78:

Assessments of inflammation and necrosis exhibited the highest levels of consensus, with at least 50% of the HPCs receiving unanimous agreement in these categories. For epithelioid growth patterns, we also observed a relatively high level of full agreement. However, for spindle architecture in our non-epithelioid clusters (orderly/less orderly/disorderly), agreement among the pathologists was lower, perhaps reflecting the subjectivity of this measure. Across 47 HPCs with 3 raters, Fleiss' Kappa scores (reported in the last row of Figure 2c - with variable category definitions per component) for individual histopathological components ranged from 0.2 to 0.6, indicating fair to moderate agreement based on the interpretation scale proposed by [18]. This degree of agreement is in line with kappa scores for several diagnostic tasks in mesothelioma [19]

The formal definition and computational procedure for Fleiss' Kappa are now provided in the Methods on page 7, line 309:

To assess agreement in our multi-centre annotation process, we used majority voting among the three expert pathologists who annotated the HPCs. Instances of unanimous agreement, where all three pathologists selected the same category and are marked with an asterisk (*). In contrast, cases of complete disagreement, where each pathologist chose a different category, are highlighted in grey in Figure 2c.

Also, inter-rater reliability was assessed using Fleiss' Kappa. For each HPC i ($N = 47$), we counted the number of raters ($n = 3$) assigning it to each category j , and computed the marginal probability of category j as:

$$p_j = \frac{1}{Nn} \sum_{i=1}^N n_{ij}, \quad 1 = \sum_{j=1}^k p_j$$

We then calculated the average proportion of agreeing rater-pairs across clusters (observed agreement \bar{P}), estimated the agreement expected by chance and scaled the excess agreement relative to the maximum possible beyond chance:

$$\kappa = \frac{\bar{P} - \bar{P}_e}{1 - \bar{P}_e}$$

A Kappa of 1 indicates perfect concordance, 0 reflects agreement no better than random, and values ≤ 0 denote worse-than-chance agreement. It is important to note that the number of categories available for annotation (denoted as j) varied across the different histomorphological components we selected, such as inflammation, necrosis, etc. As a result, the Fleiss' Kappa values are inherently influenced by this variability and are not directly comparable across components. Specifically, components with more annotation categories introduce greater choice complexity, which tends to lower agreement scores. To prevent misinterpretation, we recommend referring to the majority voting results and the asterisk indicators of full agreement as complementary measures of reliability.

4. HPC proportions as a predictor of subtypes and outcomes. A good baseline would be a MIL model (e.g. CLAM) just acting on tiles of the resections. What would be the predictive accuracy of such a model? It would motivate the *need* to use HPC. The interpretation comes from the pathologists labelling anyway, so technically interpretation of the highly attended MIL tiles would result in the same thing. I see the authors did this for the TMA cores - but that section is poorly explained and it's not clear what the bag of instances are for the MIL model.

We thank the reviewer's point regarding MIL models. To clarify our methodological choices, our approach prioritises *knowledge generation* and *interpretability* in a domain where understanding morphological subtypes has immediate clinical relevance, rather than just focusing on predictive performance metrics in mesothelioma. We aim to propose a mesothelioma-specific framework that is both label-free and discovery-driven, and which describes meaningful, permanent underlying features of the morphological landscape (the HPCs). While Multiple Instance Learning (MIL) attention mechanisms can identify tiles predictive of known labels, they are inherently limited to rediscovering patterns already annotated by pathologists. Regardless of model complexity or sampling strategy, their scope remains confined to predefined categories (e.g. Sarcomatoid, Epithelioid, and Biphasic) and they cannot capture previously unrecognised biological variations, such as sample-specific inflammation levels. By curating and isolating HPCs, our approach enables the exploration of *new* morphologies, and facilitates the analysis of their association with clinically relevant outcomes, offering deeper biological insights into mesothelioma beyond existing labels. The identification of HPCs representing previously unthought of or undescribed morphologies might be crucial both to our understanding of the disease, and to future human interpretation of histopathology. The areas of attention in HPL are driven solely by differences in histomorphology rather than human annotations, favouring true, potentially novel knowledge generation. We annotate clusters only to *boost* interpretability, not to constrain discovery, and pathologists then validate and refine the hypotheses our model produces. This is exactly the collaborative loop we envision. Even when utilising only the top ten HPCs, HPL offers a significantly more granular decomposition of histological patterns compared to MIL, enhancing robustness on unseen clinical samples. As illustrated in the figure below, the MIL attention heatmap distinguishes between tiles but fails to provide meaningful insights into the composition or

the biological prominence of these regions. In contrast, HPL enables a detailed, interpretable breakdown. Even for example, by increasing the clustering resolution, we can further dissect broad tissue areas, such as the large blue overlay (HPC15) in the figure below, and offer a clearer understanding of the tissue heterogeneity.

However, for comparison, we now extracted the same ResNet-derived feature vectors used in HPL main pipeline and trained the CLAM framework on tile features. Our simple logistic-regression and Cox model using only HPC frequency features achieve predictive performance comparable to (and outperform) MIL (subtype and survival task results below), while ensuring high reproducibility. Two survival and subtype task comparison tables are provided in the Supplementary file, and the full implementation is available on GitHub.

Leiden Resolution	Dataset	Balanced Accuracy	AUC	Sensitivity	Specificity
2.0	LATTICe-M	0.79 ± 0.05	0.88 ± 0.04	0.81 ± 0.1	0.77 ± 0.09
4.0		0.79 ± 0.04	0.89 ± 0.03	0.81 ± 0.1	0.77 ± 0.06
7.0		0.79 ± 0.04	0.88 ± 0.04	0.78 ± 0.06	0.79 ± 0.08
CLAM		0.79 ± 0.03	0.87 ± 0.04	0.77 ± 0.09	0.80 ± 0.05
2.0	TCGA-MESO	0.71 ± 0.03	0.8 ± 0.03	0.63 ± 0.06	0.79 ± 0.04
4.0		0.78 ± 0.05	0.85 ± 0.02	0.7 ± 0.05	0.85 ± 0.06
7.0		0.78 ± 0.05	0.85 ± 0.04	0.67 ± 0.08	0.89 ± 0.04
CLAM		0.67 ± 0.05	0.74 ± 0.01	0.55 ± 0.04	0.79 ± 0.03

Supplementary Table 2: Subtype classification performance metrics across clustering resolutions and datasets. Balanced Accuracy, Area under the curve (AUC), Sensitivity, Specificity scores for the mesothelioma subtype classification task on LATTICe-M and TCGA-MESO cohorts across different Leiden clustering resolutions and CLAM MIL method. The scores are averaged over 5-fold cross-validation and after sampling using the edited nearest neighbour model (ENN).

Leiden Resolutions	HPCs			Clinical and HPCs	
	Train	Test	TCGA	Train	Test
2.0	0.67 ± 0.0	0.65 ± 0.03	0.65 ± 0.01	0.68 ± 0.01	0.66 ± 0.03
4.0	0.69 ± 0.0	0.66 ± 0.03	0.66 ± 0.02	0.7 ± 0.0	0.66 ± 0.03
7.0	0.7 ± 0.01	0.65 ± 0.04	0.66 ± 0.01	0.71 ± 0.01	0.66 ± 0.04
	Risk Score			Clinical and Risk Score	
CLAM	0.60 ± 0.01	0.60 ± 0.04	0.61 ± 0.0	0.62 ± 0.02	0.62 ± 0.05

Supplementary Table 3: Concordance indices for patient outcome prediction across datasets and clustering resolutions. Patient outcome prediction was performed using the Cox proportional hazards model, reporting concordance indices (C-indices) across train, test (LATTICe-M), and external validation (TCGA) datasets at different Leiden clustering resolutions, with 5-fold cross-validation. For the primary dataset, clinical variables (including subtype, TNM stage, and age) were integrated into the patient vectors derived from HPL. Additionally, survival analysis was independently conducted using the CLAM MIL-based approach, where risk scores were generated via gated attention mechanisms.

The below text has been added to the Methods section on page 9, line 361:

We additionally benchmarked the CLAM (Clustering-constrained Attention Multiple Instance Learning)[28] framework using 128-dimensional tile embeddings extracted from our Barlow Twins-trained ResNet. Subtype classification was performed using a linear layer on top of CLAM outputs, while survival prediction was based on risk scores generated by the network and evaluated via a Cox proportional hazards model, enabling a fair comparison with HPL-based survival predictions. CLAM was trained for 50 epochs with early stopping, using the Adam optimiser with a binary loss and a learning rate of 10^{-4} . The total loss combined slide and instance-level objectives with coefficients $c_1 = 0.9$ and $c_2 = 0.3$, as follows:

$$Loss_{total} = c_1 * Loss_{slide} + c_2 * Loss_{tile}$$

The number of clusters was fixed at 8, consistent with the original CLAM configuration. WSIs were treated as bags, with subtype labels assigned at the bag level, and a gated attention, as mentioned earlier, was used to compute instance-level attention. Five-fold cross-validation, aligned with the HPL evaluation, was applied throughout.

And as a result we added this text to the discussion section on page 5, line 211:

CLAM [28] was benchmarked against HPL on both the TCGA and LATTICE-M datasets (full results in Supplementary Data). HPL consistently outperformed CLAM in both subtype classification and survival prediction, while maintaining high interpretability and biological relevance.

We have also revised the TMA section to explicitly clarify that the "bag of instances" refers to each TMA core, with the instances being the individual tiles within that core (see "Validation on Tiny Tissue Fragments", page 5, line 196):

Each TMA core was treated as a bag of instances, where the instances are individual tile embeddings extracted from the core. Using an attention-based multiple instance learning approach, we obtained a core-level representation by computing a weighted average of tile embeddings. We then performed subtype classification of each core using the core-level labels available for the TMA dataset.

The Methods section has been expanded on page 8, line 352:

In our weakly-supervised MIL setting, each TMA core is treated as a bag $B = \{h_1, h_2, \dots, h_k\}$ of k tile embeddings (Instances). Each tile h_k is obtained from our HPL ResNet-128 encoder trained using the Barlow Twins framework on the LATTICE-M dataset. To derive a representation for the entire core, we use attention-based pooling:

$$\mathbf{z} = \sum_{k=1}^K a_k \mathbf{h}_k,$$

where the attention weight a_k is computed as:

$$a_k = \frac{\exp\{\mathbf{w}^\top \tanh(\mathbf{V}\mathbf{h}_k^\top)\}}{\sum_{j=1}^K \exp\{\mathbf{w}^\top \tanh(\mathbf{V}\mathbf{h}_j^\top)\}},$$

This allows the model to learn which tiles are more informative for core-level prediction. The resulting representation z is passed to a linear classifier for subtype classification.